# Which Layer is Learning Faster? A Systematic Exploration of Layer-wise Convergence Rate for Deep Neural Networks

**Yixiong Chen**[1]     **Alan Yuille**[2]     **Zongwei Zhou**[2]
[1]The Chinese University of Hong Kong - Shenzhen     [2]Johns Hopkins University
yixiongchen@link.cuhk.edu.cn     ayuille1@jhu.edu     zzhou82@jh.edu

## Abstract

The deeply hierarchical structures enable deep neural networks (DNNs) to fit extremely complex target functions. However, the complex interaction between layers also makes the learning process of a particular layer poorly understood. This work demonstrates that the shallower layers of DNNs tend to converge faster than the deeper layers. We call this phenomenon *Layer Convergence Bias*. We also uncover the fundamental reason behind this phenomenon: Flatter local minima of shallower layers make their gradients more stable and predictive, allowing for faster training. Another surprising result is that the shallower layers tend to learn the low-frequency components of the target function, while the deeper layers usually learn the high-frequency components. It is consistent with the recent discovery that DNNs learn lower frequency objects faster.

## 1 Introduction

Over the last decade, breakthrough progress has been made by deep neural networks (DNNs) on a wide range of complicated tasks in computer vision (Krizhevsky et al., 2017), natural language processing (Sutskever et al., 2014), speech recognition (Graves et al., 2013), game playing (Silver et al., 2016), and biomedical prediction (Jumper et al., 2021). Such progress hinged on a number of advances in hardware technology, dataset construction, and model architectural designs. Among them, the invention and application of very-deep network architectures play a decisive role.

Deepening the network is an effective way to empower its fitting ability. Extensive studies (Delalleau & Bengio, 2011; Eldan & Shamir, 2016; Lu et al., 2017) compared the power between deeper and wider neural networks and showed that the polynomial growth of depth has a similar effect to the exponential growth of width. Therefore, modern DNNs (Simonyan & Zisserman, 2014; He et al., 2016) usually contain tens of layers to ensure their modeling abilities for real-world applications.

Although the practical success of deep architectures is indisputable, they make the learning hardly predictable since complex interaction happens between layers when co-adapting to the target (Yosinski et al., 2014). By now, we still have a poor understanding of how different layers learn differently. Currently, a widely accepted view relates to the vanishing gradient problem Hochreiter (1991); Hochreiter et al. (2001). The gradients are getting weaker and weaker as they move back through the hidden layers, making the shallower layers converge more slowly (Nielsen, 2015). Informally, it is reasonable that larger gradient values bring higher learning speed.

Even though this view somewhat makes sense, we seem to have little concrete evidence supporting it. In particular, it is dubious how higher-level features can be built based on the unstable features extracted by the unconverged shallower layers (Raghu et al., 2017). This paper aims to find a credible answer for the parameters of which layer are learning faster towards the convergence point (defined as the convergence rate in this work) with a systematic exploration. Our results lead to somewhat startling discoveries.

**Our Contributions.** Our point of start is illustrating that there does not seem to be a reliable positive correlation between the gradient magnitude and the convergence rate of a particular layer.

Instead, we find that shallower layers tend to converge faster than the deeper ones, even with smaller gradients. The phenomenon is called *layer convergence bias* in this paper.

We then turn our attention to excavating the underlying mechanism for the faster convergence of shallower layers. Specifically, we find out that the depth of a layer has a fundamental effect on its training: the parameters of shallower layers are usually optimized on flatter landscapes than deeper layers. This finding reveals that the gradients of shallower layers may be more predictive and thus have the potential to allow the larger learning rates (LRs) to be performed, making the convergence faster.

Finally, we find that the layer convergence bias is also tied to the frequency of the function they are modeling. When fitting a complex target function, the shallower layers tend to fit the low-frequency (usually simpler) components. On the contrary, the deeper layers struggle to fit the remaining high-frequency components. It is a consistent result of the recent discovery that DNNs prioritize learning low-frequency components of the modeling function, while having very low learning speed on high-frequency components that tend to be more complex (Rahaman et al., 2019). This finding provides us with another perspective to understand why deeper layers learn more slowly.

We believe that understanding the roots of such a fundamental convergence bias can give us a better grasp of the complicated learning process of DNNs. In turn, it can motivate more in-depth algorithmic progress for the deep learning community.

This paper is organized as follows. In Section 2, we introduce our method for measuring convergence speed for different layers, and formally define the layer convergence bias. In Section 3, we examine the relationship between gradient magnitude and convergence rate, and show that the shallower layers tend to converge faster even with smaller gradients. Then in Section 4, we analyze the mechanism behind the layer convergence bias in DNN training. The layer-frequency correspondence is demonstrated in Section 5. The practical significance of layer convergence bias is presented in Section 6. We further discuss the related work in Section 7 and conclude in Section 8.

## 2    LAYER CONVERGENCE BIAS

The deep architecture of DNNs is arguably one of the most important factors for their powerful fitting abilities. With the benefit brought by the deep structures, there are also extra complexities in the training process coming into being. So far, we do not have a firm conclusion about whether some layers are learning faster than others.

For examining the convergence progress for a DNN, a common practice is checking its loss curve. However, this is not applicable for comparing the convergence between different layers. In this work, we define a measurement for layer-wise convergence in the following.

**Definition 2.1** (Layer-wise convergence rate) *At the training time $t$, let the deep neural network with $L$ layers $\{T_l^{(t)}\}_{l=1}^{L}$ be $f(\boldsymbol{x}) = (T_L^{(t)} \circ T_{L-1}^{(t)} \circ \cdots \circ T_1^{(t)})(\boldsymbol{x}) : \mathbb{R}^i \to \mathbb{R}^o$, where $i, o$ are the dimension of its inputs and outputs. We use $\theta_l^{(t)}$ to denote the parameters of the l-th layer $T_l^{(t)}$. Assuming that $\theta_l^{(t)}$ can finally converge to its optimal point $\theta_l^*$ when $t \to \infty$, we define the convergence rate of $\theta_l$ during the time interval $[t_1, t_2]$ to be*

$$C_l^{(t_1, t_2)} = \frac{1}{(t_2 - t_1)} \cdot \frac{\|\theta_l^{(t_1)} - \theta_l^*\|_2 - \|\theta_l^{(t_2)} - \theta_l^*\|_2}{\|\theta_l^{(t_0)} - \theta_l^*\|_2},$$

*where $t_0$ denotes the time point when the training starts.*

In this definition, the numerator $\|\theta_l^{(t_1)} - \theta_l^*\|_2 - \|\theta_l^{(t_2)} - \theta_l^*\|_2$ denotes how much the distance of the parameter $\theta_l$ to the optimal point is shortened in the period $[t_1, t_2]$. The denominator $\|\theta_l^{(t_0)} - \theta_l^*\|_2$ represents the distance between the initial point to the convergence point, whose primary function is to normalize the speed, allowing the convergence of different layers to compare with each other. Thus, the convergence rate of $\theta_l$ can be understood as the ratio of normalized distance to time. Common optimization works (Yi et al., 1999; Nesterov, 2003) defined the rate of convergence for $\theta$ as $\lim_{k \to \infty} \frac{\|\theta^{(k+1)} - \theta^*\|_2}{\|\theta^{(k)} - \theta^*\|_2}$. It focuses on measuring an exponential level convergence when the

optimization step goes to infinity. Since the difference in convergence rates between layers usually appears at an early stage of training, and it is not large enough to compare at an exponential level, we define our new convergence metric to present the convergence difference in a clearer way.

**Observation 2.1** (Layer convergence bias). *For $l_1 < l_2$, $\exists \tilde{t} > 0$, such that $C_{l_1}^{(t_1, t_2)} > C_{l_2}^{(t_1, t_2)}$ when $t_1 < t_2 < \tilde{t}$.*

Layer convergence bias indicates that at an early training phase $t < \tilde{t}$, the parameters $\theta_{l_1}$ of a shallower layer $l_1$ tend to move to $\theta_{l_1}^*$ faster than a deeper layer $\theta_{l_2}$ moving to $\theta_{l_2}^*$. In the following, we use both synthetic and real datasets to show that the layer convergence bias appears for both fully-connected neural networks (FCNNs) and convolutional neural networks (CNNs).

# 3 VERIFICATION OF LAYER CONVERGENCE BIAS

In this section, we try to substantiate the central claim of this work. First, we use the FCNNs to show that the shallower layers tend to converge faster than the deeper layers on the regression task, even when the gradient values for shallower layers are smaller. We then use CNNs with modern architectures to verify that layer convergence bias is a common phenomenon in practical applications. All experimental settings in this work can be found in Appendix A.1.

## 3.1 LAYER CONVERGENCE BIAS IN FULLY-CONNECTED NETWORKS

For FCNNs, we construct a simple regression task to demonstrate that layers with smaller gradients do not necessarily learn more slowly than layers with larger gradients. The fitting target is $f(x) = sin(x) + \frac{1}{3}sin(3x) + \frac{1}{10}sin(10x) + \frac{1}{30}sin(30x)$, with mean square error loss for training.

First, we use the FCNN [1-32-32-32-1] with the Sigmoid activations as a simple example. In the following analysis, the first fully-connected layer (1-32) is named *Layer 1*, and the subsequent two layers (32-32) are called *Hidden layer 1, Hidden layer 2* respectively. The gradient values and the convergence processes for these layers are shown in Fig. 1 (a). Two observations can be obtained from the plots: 1) The gradient of *Hidden layer 1* is nearly always smaller than the gradient of *Hidden layer 2*. 2) Although shallower layers have smaller gradients, they seem to converge faster. For the first 50 epochs, the shallower layers are moving faster to their convergence point (*e.g.*, $C_{Layer\ 1}^{(t_0, t_{50})} \approx 0.012$, $C_{Hidden\ layer\ 1}^{(t_0, t_{50})} \approx 0.009$, $C_{Hidden\ layer\ 2}^{(t_0, t_{50})} \approx 0.006$), which is inconsistent with the previous view that higher gradients lead to faster learning (Nielsen, 2015).

To further validate the above results with a deeper network, we adopt residual connections (He et al., 2016) for the FCNN (deep network fails to be trained in this task without residual connections) and use the ReLU activation function. The FCNN [1-(128-128)-(128-128)-(128-128)-(128-128)-1] with four residual blocks of width 128 shows similar results to the shallow FCNN without residual connection (see Fig. 1 (b)). In this case, the difference in layer-wise convergence rate can be observed even earlier (*i.e.*, $C_{Res-Block\ 1}^{(t_0, t_5)} \approx 2C_{Res-Block\ 4}^{(t_0, t_5)}$), which shows that the layer convergence bias also happens for deeper FCNNs with residual connections. It is noteworthy that our convergence metric is crucial to observe the layer convergence bias, which is elaborated in Appendix A.2.

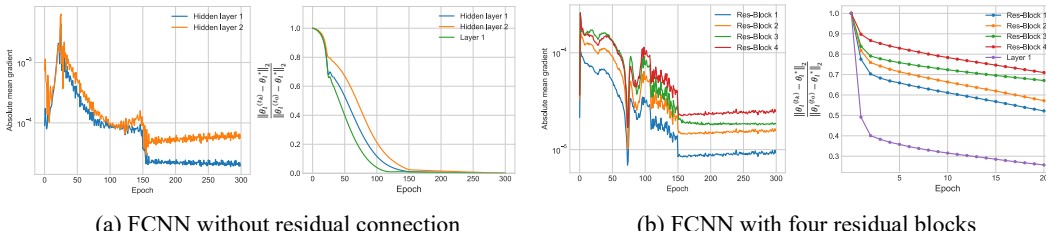

(a) FCNN without residual connection      (b) FCNN with four residual blocks

Figure 1: Left (a,b): The absolute mean gradient values for different layers for FCNNs w/o residual connections in training. For both networks, deeper layers have larger gradients. Right (a,b): The convergence process of different layers for FCNNs. Shallower layers converge faster.

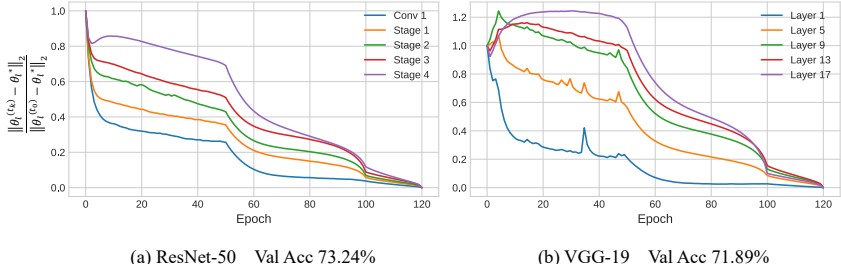

(a) ResNet-50    Val Acc 73.24%          (b) VGG-19    Val Acc 71.89%

Figure 2: The convergence process of ResNet-50 and VGG-19 on ImageNet. During the first 50 epochs, shallower layers converge much faster than deeper layers. After the learning rate decays at the 50th epoch, parameters of deeper layers accelerate to move to their convergence points.

Clearly, these results cannot reconcile with the previous view that larger gradients bring a higher learning speed for deeper layers, at least for the DNNs used in this work. Instead, from the optimization point of view, the parameters of shallower layers are learning faster to converge.

## 3.2 LAYER CONVERGENCE BIAS IN CONVOLUTIONAL NETWORKS

Real-world datasets are very different from the synthetic data used in our previous experiments. In order to utilize the layer convergence bias to understand and better improve DNNs in real applications, it is important to verify whether the layer convergence bias holds for CNNs on images.

In the following experiments, we examine the layer-wise convergence process on ImageNet (Russakovsky et al., 2015) dataset with both ResNet-50 (He et al., 2016) and VGG-19 (Simonyan & Zisserman, 2014). We train the CNNs for 120 epochs with learning rate decay at the 50th epoch ($0.1 \rightarrow 0.01$) and the 100th epoch ($0.01 \rightarrow 0.001$). The training processes are shown in Fig. 2.

For ResNet-50, we visualize the learning process of the first convolutional layer and its subsequent four stages. One can easily observe that at the beginning of training, the shallower layers converge much faster than the deeper layers ($C_{Stage\ 1}^{(t_0, t_{20})} \approx 3 C_{Stage\ 4}^{(t_0, t_{20})}$). However, after the learning rate decays at the 50th epoch, deeper layers begin to learn effectively and achieve a higher convergence rate than the shallower layers ($C_{Stage\ 1}^{(t_{50}, t_{60})} \approx 0.5 C_{Stage\ 4}^{(t_{50}, t_{60})}$). We conjecture that the initial learning rate is too large for the deeper layers to learn.

For VGG-19, we visualize its 1st, 5th, 9th, 13th, and 17th layers. This network show a more significant convergence difference between layers than ResNet-50. At the first training stage with the initial learning rate, $\|\theta_l^{(t_5)} - \theta_l^*\| > \|\theta_l^{(t_0)} - \theta_l^*\|$ for $l \in \{5, 9, 13, 17\}$, which means that all layers but the first one even slightly diverge. Usually, the divergence appears when the learning rate is too large. This phenomenon confirms that the deeper layers cannot effectively learn with the large learning rate at the beginning.

The experiments of FCNNs and CNNs verify that layer convergence bias is a common phenomenon for DNNs. In Section 5 and Appendix A.3, A.4, we discuss the factors that would affect the phenomenon, and some in-depth findings they reveal.

## 4 MECHANISM BEHIND LAYER CONVERGENCE BIAS

So far, our investigation shows that the seemingly-right perspective for linking the layer-wise gradient and convergence rate is tenuous, at best. Both FCNNs and CNNs demonstrate an evident bias that shallower layers learn faster. *Can we explain why this is the case?*

**Gradient Predictiveness.** Since gradient values cannot determine the convergence rate, we wonder if the directions of the gradients play a more critical role. More chaotic update directions make convergence slower. Here we examine the gradient predictiveness (Santurkar et al., 2018) of different layers. If the gradient behavior is "predictive", less change in the gradient directions would appear when 1) the gradients are calculated with different batches of data; 2) the parameters of other layers update. Predictiveness can also be simply understood as the stability of gradient direction.

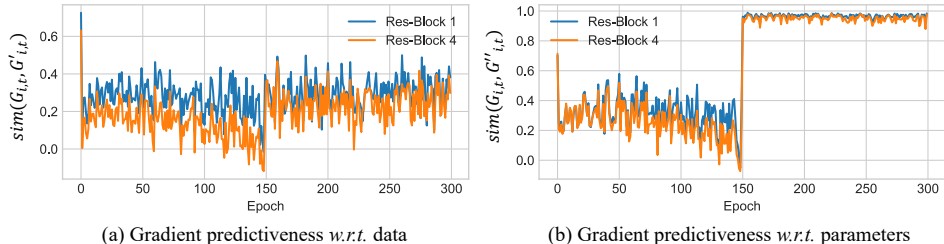

(a) Gradient predictiveness *w.r.t.* data

(b) Gradient predictiveness *w.r.t.* parameters

Figure 3: The gradient predictiveness of shallower and deeper layers of FCNN. The learning rate decreases from 0.1 to 0.01 at Epoch 150.

**Definition 4.1** *Let $(x^{(t)}, y^{(t)})$ be a batch of input-label pairs for the DNN to train at time $t$, and $(x'^{(t)}, y'^{(t)})$ be another batch of data. We define the gradient predictiveness of the lth layer at time $t$ w.r.t. data be the cosine similarity $sim(G_{l,t}, G'_{l,t}) = \frac{\|G_{l,t}G'_{l,t}\|}{\|G_{l,t}\|\|G'_{l,t}\|} \in [-1, 1]$. Likewise, the gradient predictiveness w.r.t. parameters is defined as $sim(G_{l,t}, G''_{l,t})$, where*

$$G_{l,t} = \nabla_{\theta_l^{(t)}} L(\theta_1^{(t)}, ..., \theta_L^{(t)}; x^{(t)}, y^{(t)})$$

$$G'_{l,t} = \nabla_{\theta_l^{(t)}} L(\theta_1^{(t)}, ..., \theta_L^{(t)}; x'^{(t)}, y'^{(t)})$$

$$G''_{l,t} = \nabla_{\theta_l^{(t)}} L(\theta_1^{(t+1)}, ..., \theta_{l-1}^{(t+1)}, \theta_l^{(t)}, \theta_{l+1}^{(t+1)}, ..., \theta_L^{(t+1)}; x^{(t)}, y^{(t)})$$

Here, $G_{l,t}$ corresponds to the gradient of $\theta_l^{(t)}$. $G'_{l,t}$ is the gradient of this layer with another batch of data, while $G''_{l,t}$ means the gradient after all the other layers have updated to new values. Therefore, $sim(G_{l,t}, G'_{l,t})$ indicates the stability of gradients with different data batches. $sim(G_{l,t}, G''_{l,t})$ reflects whether the currently estimated gradient is in a consistent decreasing direction when the loss landscape is affected by the updating of other layers' parameters. The gradient predictiveness during training is shown in Fig. 3, where Res-Block 1 has more predictive gradients than Res-Block 4.

**Visualizing the Loss Landscapes.** We are curious about why gradients for deeper layers have poorer predictiveness. A hypothesis is that the loss landscapes for deeper layers are more rugged, making the parameters fluctuate more. A straightforward method to validate this hypothesis is plotting the loss landscapes for the parameters. To do this for a particular layer $l$, one can choose a central point $\theta_l^*$ and two direction vectors $d_{l,1}$, $d_{l,2}$. Then the loss landscape can be drawn with

$$f(\beta_1, \beta_2) = L(\theta_l^* + \beta_1 d_{l,1} + \beta_2 d_{l,2})$$

in the 3D space with $\beta_1, \beta_2$ forming a simplified parameter space. In this work, we generate random Gaussian directions for different layers, and normalize them to obtain the same norm of the corresponding layer. Specifically, we make the replacement $d_l \leftarrow \frac{d_l}{\|d_l\|} \|\theta_l^*\|$ for a fully connected layer. For a convolutional layer, we use filter-wise normalization $d_l^k \leftarrow \frac{d_l^k}{\|d_l^k\|} \|\theta_l^{k*}\|$ as in (Li et al., 2018), where $d_l^k$ represents the $k$th filter of the $l$th layer. We set both $\beta_1$ and $\beta_2$ in the domain of $[-1, 1]$.

**Landscapes for FCNN.** The loss landscapes for four residual blocks of the FCNN are shown in Fig. 4. For the shallower blocks, the surfaces are flatter near the minimizer, meaning that the gradient magnitudes may be small. However, small gradients do not necessarily lead to slow learning speed in this case. Combined with the gradient predictiveness discussed above, a flatter loss landscape may lead to more consistent gradient directions, making the learning more smooth.

**Landscapes for CNNs.** The loss landscapes for ResNet-50 and VGG-19 on ImageNet are shown in Fig. 5. It is interesting that deep convolutional networks with/without residual connections present totally different loss landscapes. For ResNet-50, its landscapes near the convergence point $\theta_l^*$ are smooth and nearly convex, making the neural network easier to train. On the contrary, VGG-19 has much more shattered landscapes, the initial iterations probably lie in the chaotic regions, prohibiting its training (Balduzzi et al., 2017). This may explain the much less efficient convergence towards the optimal point for VGG than ResNet at the initial phase (Fig. 2).

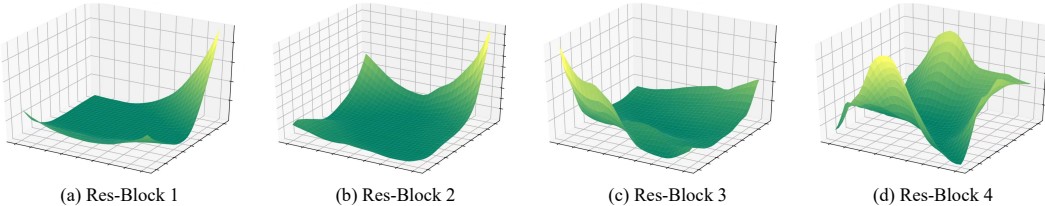

(a) Res-Block 1    (b) Res-Block 2    (c) Res-Block 3    (d) Res-Block 4

Figure 4: The loss landscapes of different layers of FCNN. Deeper layers are optimized on more rugged landscapes, slowing down the learning process.

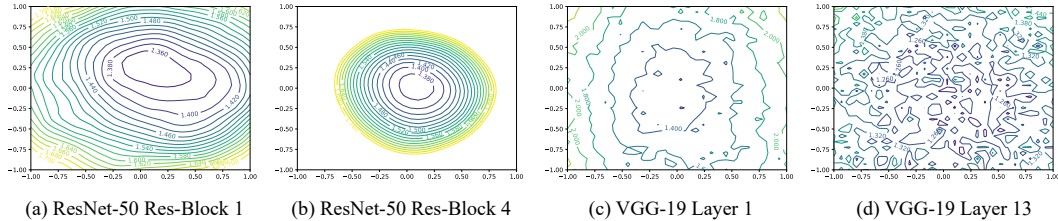

(a) ResNet-50 Res-Block 1  (b) ResNet-50 Res-Block 4  (c) VGG-19 Layer 1  (d) VGG-19 Layer 13

Figure 5: The loss landscapes of different layers of ResNet-50 (a,b) and VGG-19 (c,d) on ImageNet. The shallower layers for both networks have flatter minima, making them converge faster than the deeper layers. The plots for all layers can be found in Appendix A.5.

Comparing different layers in the CNNs, the answer for layer convergence bias becomes clearer. The key difference between different layers' loss landscapes of ResNet-50 is the sharpness of the local minima (Fig. 5 (a,b)). We conjecture it is because of a well-known fact that the shallower layers of CNNs tend to learn general features which are applicable to various datasets and tasks, while the deeper layers usually learn task-specific features (Yosinski et al., 2014). Before our work, (Zeiler & Fergus, 2014) also revealed that the general features in a five-layer CNN stabilized faster than the specific features. Since the general features are more evenly distributed, they usually cause less fluctuation for training, leading to flatter optima. Theoretically, flatter minimizers are easier to be found by SGD optimizers (Pan et al., 2020). For VGG-19, its shallower and deeper layers also have flatter and sharper minima (Fig. 5 (c,d)), respectively. The shattered loss landscape for its deeper layers may also explain its inefficient learning process with a large learning rate (Fig. 2 (b)).

Here we summarize the mechanism behind layer convergence bias: the parameters of shallower layers are easier to optimize due to their flatter loss landscapes. At a higher level, shallower layers learn general features, which are usually easier.

## 5 DEEPER LAYERS FIT THE HIGH-FREQUENCY COMPONENTS

Recent advances in the learning process of DNNs (Rahaman et al., 2019; Ronen et al., 2019; Xu & Zhou, 2021) revealed that the low-frequency components of the target function are fitted much faster than the high-frequency components. There is a natural question about whether there is some inherent link between layer convergence bias and this result. In this section, we investigate the answer, and surprisingly find that: the low-frequency parts are usually fitted by the shallower layers, while the remaining higher frequencies are mainly learned by the deeper layers. It provides us with an alternative perspective to understand the layer convergence bias.

**The Correspondence for FCNN.** With the residual structures, we can straightforwardly visualize what each block of a FCNN learns. Considering the FCNN with one input layer $z_0 = T_0(x) : \mathbb{R}^1 \to \mathbb{R}^{128}$, four residual blocks $z_l = T'_l(z_{l-1}) = T_l(z_{l-1}) + z_{l-1} : \mathbb{R}^{128} \to \mathbb{R}^{128}$, $l \in \{1, 2, 3, 4\}$, and an output layer $y = T_5(z_4) : \mathbb{R}^{128} \to \mathbb{R}^1$. The whole network can be expressed as

$$y = T_5(z_1 + T_2(z_1) + T_3(z_2) + T_4(z_3)) = T_5(z_1) + T_5(T_2(z_1)) + T_5(T_3(z_2)) + T_5(T_4(z_3))$$

if the output layer $T_5$ is a linear transformation. The fitting results for each layer are shown in Fig. 6. It can be seen that the deeper layers tend to fit the more complex components of the target function $y = sin(x) + \frac{1}{3}sin(3x) + \frac{1}{10}sin(10x) + \frac{1}{30}sin(30x)$. Besides the curvature, the fitted functions

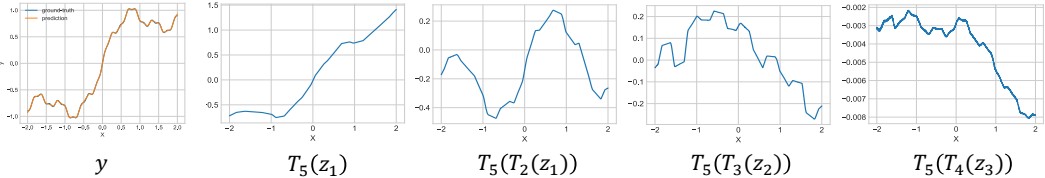

$y$ $\quad\quad T_5(z_1)$ $\quad\quad T_5(T_2(z_1))$ $\quad\quad T_5(T_3(z_2))$ $\quad\quad T_5(T_4(z_3))$

Figure 6: The visualization of what each residual block of the FCNN learns. From the first to the fourth block, the fitted function becomes more complex with smaller amplitude.

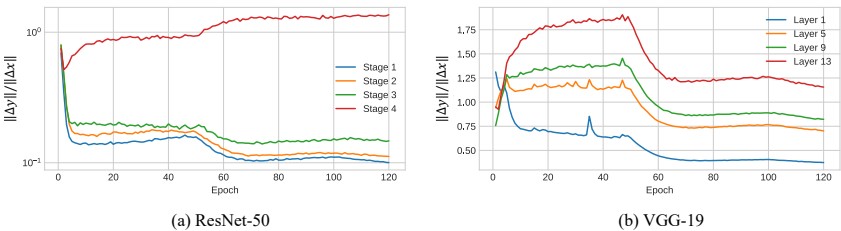

(a) ResNet-50 $\quad\quad\quad\quad\quad\quad\quad\quad\quad\quad\quad\quad$ (b) VGG-19

Figure 7: The visualization of response frequencies for CNNs. As the training goes on, deeper layers become more sensitive to perturbations, indicating that they have higher response frequencies.

are also consistent with the amplitudes of the components. Specifically, the ranges of the four fitted functions are $2.3, 0.7, 0.5$, and $0.06$, which are similar to the four components. This result further confirms the relationship between layers and frequencies.

**The Correspondence for CNNs.** For CNNs, we verify their layer-frequency correspondence through the response frequency (Xu et al., 2019). In a nutshell, if an input-output mapping $f$ possesses significant high frequencies, then a small change in its input induces a large change in the output. We generate standard Gaussian-distributed input $x$ for different residual blocks of ResNet-50 and different layers of VGG-19. At the same time, small Gaussian perturbation $\Delta x$ is added to the input. A larger change $\Delta y$ of the layer output means the layer handles higher frequencies. The response frequencies are shown in Fig. 7. At the first 5 epochs of training on ImageNet, different layers for both ResNet-50 and VGG-19 do not show significantly different response frequencies. But after about ten epochs, the response frequencies for deeper layers (*e.g.*, stage 4 for ResNet-50, layer 13 for VGG-19) increase while the shallower layers show lower response frequencies. Therefore, we conclude that the layer-frequency correspondence also holds for CNNs. In addition, it is not an innate nature of the layers, but a result of the training process.

**How the target frequency affects layer convergence bias?** To demonstrate the effect of layer-frequency correspondence on the layer convergence bias, we try fitting simpler targets with less high-frequency components, and see what would happen to the layer-wise convergence rate of FCNN. In Fig. 8 (a-d), we only keep several lowest frequencies of the target, *e.g.*, the target function $y = sin(x)$ is named "Complexity=1", and $y = sin(x) + \frac{1}{3}sin(3x)$ is named "Complexity=2", *etc.* After discarding more and more high-frequency components, the deeper layers converge faster and faster. In this case, the layer convergence bias does not strictly hold anymore. In Fig. 8 (b), the Res-Block 4 converges faster than Res-Block 3 after the 5th epoch. In Fig. 8 (c), the Res-Block 4 converges with a similar speed as Res-Block 2, while the Res-Block 3 even learns faster than Res-Block 2. It seems that removing the high-frequency component that corresponds to a deep layer can effectively accelerate its training. For CNNs, we also observe similar phenomena (Fig. 8 (e-h)). On simpler targets (*e.g.*, CIFAR 10), the deeper layers converge faster than on more complex targets (*e.g.*, CIFAR100). An implication of this result is that the data complexity may be too low for the model. In practice, CIFAR datasets only need ResNet-18 to fit well (Wu et al., 2020).

In fact, (Rahaman et al., 2019) had shown that different layers have some links to different frequencies, but the authors did not provide further insight for this phenomenon. This work verifies the underlying relationship between layers and fitting frequencies, and establishes a connection for this relationship to the layer convergence bias.

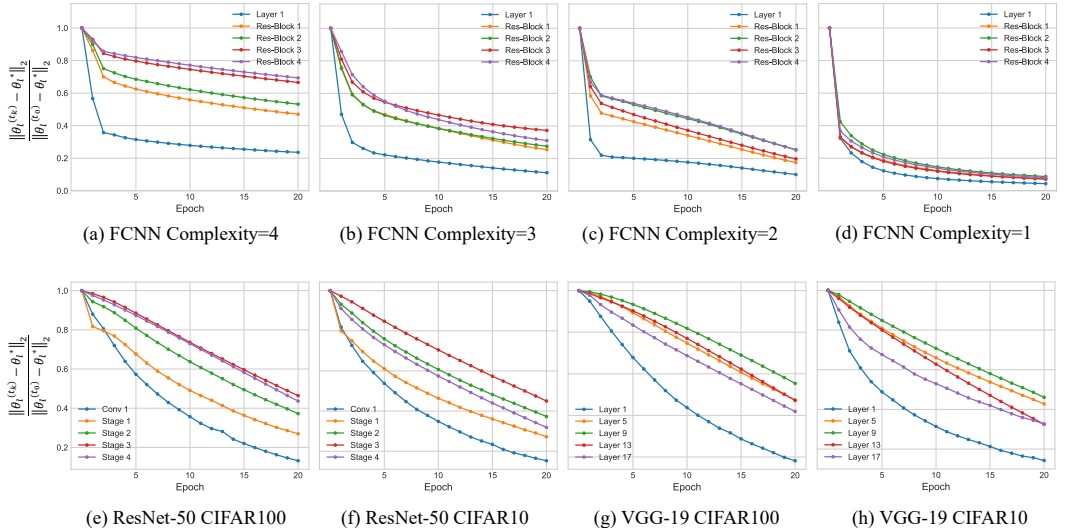

Figure 8: The convergence curves with different learning target complexities. (a-d): Decreasing target complexities for FCNNs. The deeper layers accelerate more than the shallower ones when high-frequency components are removed. (e-h): For CNNs, the deepest layers (*i.e.*, Stage 4 / Layer 17) learn faster on CIFAR10 than on CIFAR100 while the other layers do not change much.

## 6 PRACTICAL SIGNIFICANCE

Up to now, we have been analyzing the layer convergence bias from a theoretical perspective. This section discusses its practical use to drive the development of DNN architecture design, and a new explanation for the acceleration effect of transfer learning with the help of layer convergence bias.

### 6.1 DNN ARCHITECTURE DESIGN

Modern CNN architectures (He et al., 2016) usually contain layers from narrow to wide (*e.g.*, 64 channels of the first layer to 2048 channels of the last layer). From the perspective of computational complexity, the narrower shallower layers make the corresponding large feature maps less computation-consuming. Considering the layer convergence bias, deeper layers with larger capacities are also beneficial for the corresponding high-frequencies to be learned easier. Although this is a common design for CNNs, Transformers (Dosovitskiy et al., 2020) usually apply the same architecture for all encoders. For a vision Transformer with 12 encoders, we use encoders with width 2/4/8 to construct three variants. The variants only differ in the arrangement of different encoders, we use $W$ to denote the widths, and $N$ to denote the number of each kind of encoders. The configures are summarized below:

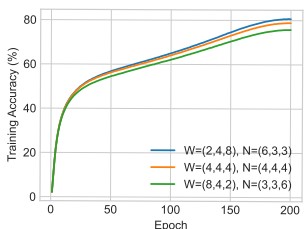

Figure 9: Performance of three variants of ViTs on ImageNet.

- deeper encoders wider: $W = (2, 4, 8)$, $N = (6, 3, 3)$
- vanilla architecture: $W = (4, 4, 4)$, $N = (4, 4, 4)$
- deeper encoders narrower: $W = (8, 4, 2)$, $N = (3, 3, 6)$

Fig. 9 shows their performances, with the best accuracy of 80.75%, 78.88%, and 75.75%, respectively. We find that with the same number of parameters, putting the wider layers deeper results in higher training performance. This finding may serve as an effective way to improve the model capacity. The causal connection between layer complexity distribution and model performance is discussed in Appendix A.6. And layer convergence bias for ViT is analyzed in Appendix A.7.

## 6.2 ACCELERATION EFFECT OF TRANSFER LEARNING

Transfer learning (fine-tuning with the pre-trained models) is a widely-used technique that can accelerate the model convergence (Shao et al., 2018b;a; Liang & Zheng, 2020). We show the layer convergence curves w/o transfer learning on the Flowers dataset (Nilsback & Zisserman, 2006). When training from scratch (Fig. 10 (a)), the shallower layers converge faster so that the deeper layers can extract semantic features based on basic features. Local minima of *Stage 4* is sharp in this case. However, with transfer learning (Fig. 10 (b)), deeper layers can directly be built on the pre-trained basic features. The *Stage 4* shows a much higher convergence rate among all layers, its loss landscape also becomes flatter. Two observations that are not consistent with layer convergence bias are summarized in the following: 1) the pre-trained shallower layers are nearly optimal, so they don't present fast convergence in transfer learning; 2) although the pre-trained deeper layers are not as optimal as the shallower layers do, their loss landscapes are much flatter than training from scratch, which makes them converge much faster.

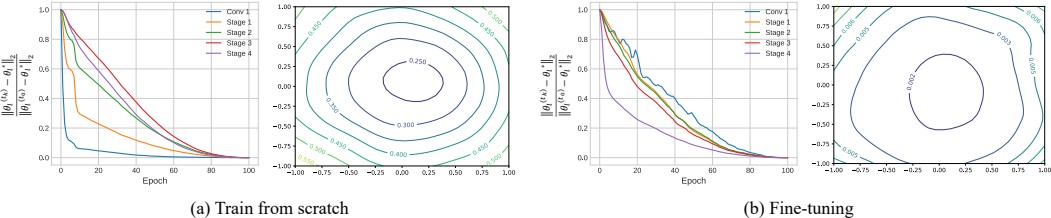

(a) Train from scratch (b) Fine-tuning

Figure 10: Effects of transfer learning on the training process. Left (a,b): The layer convergence process of ResNet-50. Right (a,b): The loss landscapes of *Stage 4* w/o transfer learning.

## 7 RELATED WORK

DNNs with gradient-based training show great potential to fit targets with arbitrary complexities (Hornik et al., 1989; Leshno et al., 1993), given sufficient width. With the advances in the last decade to verify the capability of the depth of universal approximators (Delalleau & Bengio, 2011; Eldan & Shamir, 2016; Lu et al., 2017), practitioners tried to reduce the width of neural networks by adding more layers (Simonyan & Zisserman, 2014; He et al., 2016; Huang et al., 2017). We are also inspired by research on local properties (sharpness/flatness) of loss functions at minima (Keskar et al., 2017; Li et al., 2018) and relationship between convergence rate and generalization (Hardt et al., 2016). Furthermore, LARS optimizer (You et al., 2017) shares some valuable insights on layer convergence, which are discussed in Appendix A.8. In practice, the idea of layer convergence bias had been intuitively applied to accelerate DNN training (Huang et al., 2016; Brock et al., 2017) and mitigating catastrophic forgetting (Ramasesh et al., 2020). The arrangement schemes of CNN/Transformer blocks were explored by (Liu et al., 2022b;a).

## 8 CONCLUSION

In this work, we empirically studied the phenomenon that the shallower layers of DNNs tend to converge faster than the deeper layers, called layer convergence bias. This phenomenon is a natural preference in the process of DNN training: the shallower layers are responsible for extracting low-level features which are more evenly distributed and easier to learn, while deeper layers refine these features to do specific tasks. This makes the loss landscapes for shallower layers flatter than the landscapes for deeper layers, making shallower layers converge faster. In addition, this work established a connection between layers and learned frequencies. By showing deeper layers tend to fit the high-frequency components in the target function, we can understand the layer convergence bias from another perspective. We finally took DNN architecture design and transfer learning as two examples to show how theoretical findings in this work can shed light on the practical applications of deep learning. For progress to continue, a more in-depth understanding of the properties of neural networks is needed. We also hope that the layer convergence bias can inspire more practical improvements in the DNNs' architecture design and training schemes.

## ACKNOWLEDGMENTS

This work was supported by the Lustgarten Foundation for Pancreatic Cancer Research and the McGovern Foundation.

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

# A APPENDIX

## A.1 EXPERIMENTAL SETTINGS

**Datasets.** The synthetic and real datasets are summarized in the Tab. 1

Table 1: Descriptions and statistics of the datasets used in this work.

| Dataset | Size (train/test) | Classes | Data description |
|---|---|---|---|
| Sine regression | 5000/5000 | n/a | Function with four sine components, domain [-2,2] |
| ImageNet | 1,281,167/50,000 | 1000 | Photos of common objects |
| CIFAR-10 | 50,000/10,000 | 10 | Photos of common objects, image sizes $32 \times 32$ |
| CIFAR-100 | 50,000/10,000 | 100 | Photos of common objects, image sizes $32 \times 32$ |
| Flowers | 1,088/272 | 17 | Find-grained photos of flowers |
| FGVC Aircraft | 6,667/3,333 | 100 | Find-grained photos of aircrafts |
| Caltech-101 | 3,060/6,084 | 102 | Photos/paintings/sketches of common objects |
| CUB-200 | 5,994/5,794 | 200 | Find-grained photos of birds |
| DomainNet painting | 50,416/21,850 | 345 | Oil Paintings, murals, drawings, tattoos |

**Network Architectures.** The FCNNs, CNNs, and Vision Transformers are summarized in the Tab. 2.

Table 2: Complexities and architectures of DNNs used in this work.

| Model | #Parameters | Mult-adds | Architecture description |
|---|---|---|---|
| FCNN (no res) | 2k | 10k | 4 fc layers [1-32-32-32-1] |
| FCNN (res) | 132k | 390k | fc [1-128] $\to$ 4 res-blocks [128-128-128] $\to$ fc [128-1] |
| ResNet-50 | 25.6M | 4.1G | conv $\to$ 4 stages with [3,4,6,3] res-blocks $\to$ fc |
| VGG-19 | 143.7M | 19.8G | 16 conv layers, 3 fc layers |
| ViT | 9.9M | 77.2M | 12 Transformer encoder blocks (basic width 256), 1 fc layer |

**Training Hyper-parameters.** For the regression task, we train FCNNs with SGD optimizers for 300 epochs. The initial learning rate is 0.1, with a learning rate decay (to 0.01) at the 150th epoch. The batch size is 128, no weight decay ($L_2$ regularization) is conducted.

For the ImageNet classification task with CNNs, we train ResNet-50 and VGG-19 for 120 epochs with SGD optimizers. The initial learning rate is 0.1, with learning rate decays at the 50th and 100th epoch to 0.01 and 0.001, respectively. The batch size is 256, the input image size is $224^2$, and the weight decay coefficient is $10^{-4}$.

For Vision Transformers on ImageNet dataset, we train them for 200 epochs with Adam optimizers. The peak learning rate is set to 0.0003. We use linear learning rate warm-up for 10,000 iterations, and a subsequent cosine learning rate decay. The batch size is 256, the input image size is $224^2$, and the weight decay coefficient is $10^{-4}$.

For CNN image classification on other datasets, we train models for 100 epochs with SGD optimizers. Initial learning rate of 0.01 and cosine learning rate scheduler are applied. The batch size is 128, the input image sizes are $32^2$ (for CIFAR) and $224^2$ (for Flowers, Aircraft, Caltech, CUB, and DomainNet), and the weight decay coefficient is $10^{-4}$.

## A.2 CONVERGENCE MEASUREMENT USING WEIGHT VARIATION

In Section 2, we have introduced the convergence measurement in this work. This measurement is simple and straightforward, and it can show how each layer in a DNN converges during the whole training process (Fig. 1 for fully connected networks and Fig. 2 for CNNs) by examining the distance between the training parameters and the converged parameters. However, it has not been verified whether calculating the parameter distance variation to the convergence point between two adjacent epochs is necessary. After all, the measurement highly depend on the convergence point, which can only be obtained after the whole training process.

We come up with a simplified convergence measurement. This method uses weight variation as a metric to examine how fast a layer is learning, and whether this layer reaches a state of convergence. If a layer is learning actively, it is reasonable that its weights variate drastically during training. For the converged layers, their weights usually keep stable. So we use $\|\theta_l^{(t_k)} - \theta_l^{(t_{k+1})}\|_2 / \|\theta_l^{(t_k)}\|_2$, the normalized weight variation of layer $l$ during epoch $k$ and $k + 1$, to illustrate how actively it is learning.

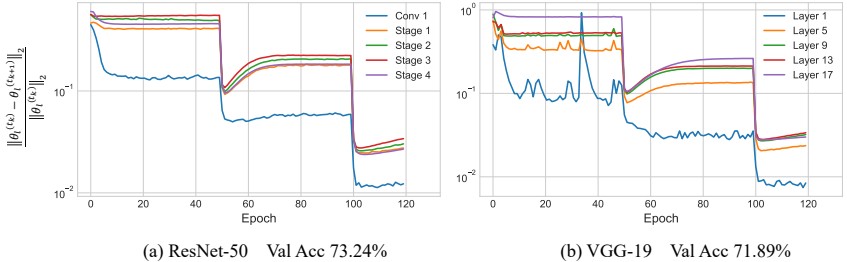

(a) ResNet-50    Val Acc 73.24%          (b) VGG-19    Val Acc 71.89%

Figure 11: The convergence processes of ResNet-50 and VGG-19 on ImageNet. The results are illustrated with weight variations. The learning rate decays at epoch 50 and epoch 100.

The results of ResNet-50 and VGG-19 training process on ImageNet are shown in Fig. 11. From this plot, we can see that after learning rate decays at epoch 50 and 100, the weight variations drop evidently. However, the weight variations of each layer do not show apparent decreasing trend when the learning rate keeps stable, which indicates that the training of DNNs do not converge as usual convex optimization problems do (*e.g.*, linear programming). Therefore, it is hard for us to compare the convergence rates of different layers by observing their convergence curves. We cannot find a clear clue like what was given by the convergence measurement in Section 2 to get the layer convergence bias. All in all, we can safely claim that, it is crucial for the convergence metric to consider direction information to measure how fast different layers are learning towards their convergence points. Our previous convergence measurement really needs to examine convergence by calculating the parameter distance between the current point to convergence point.

## A.3 FACTORS AFFECTING LAYER CONVERGENCE BIAS

In Section 5, we have shown that the complexity of the datasets is an important factor affecting layer convergence bias. When the fitting target function is complex enough with both low and high frequency components, the shallower layers learn the low low-frequency components while the deeper layers learn the high-frequency components. Here we use the FCNNs with residual connections to show whether some other important factors would affect the layer convergence bias. All following experiments are conducted on the same regression task in Section 3.

**Model Depth.** The default architecture used in previous experiments is the four-blocks FCNN, here we try adding more blocks to make the network deeper and see what change will happen. As shown in Fig. 12, all the networks show layer convergence bias. With more and more res-blocks, the overall convergence of the network becomes slightly faster.

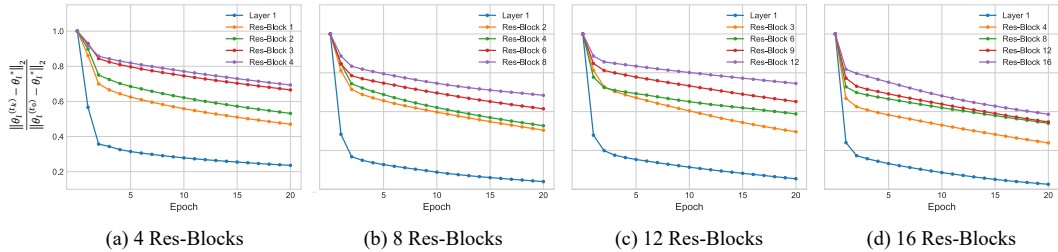

| (a) 4 Res-Blocks | (b) 8 Res-Blocks | (c) 12 Res-Blocks | (d) 16 Res-Blocks |

Figure 12: The convergence process of FCNNs with different number of res-blocks.

**Learning Rate.** The results with different learning rates are shown in Fig. 13. When the learning rate gets smaller, layer convergence bias becomes weaker. This is because the gradient predictiveness *w.r.t.* parameters of all layers get close to 1 (see Fig. 3 (b,right) for the predictiveness with the learning rate of 0.01). In this case, a layer is less influenced by the updates of parameters in other layers, only the gradient predictiveness *w.r.t.* data matters for the convergence rate. In addition, smaller learning rates are beneficial for the deeper layers to converge because of their sharper minima.

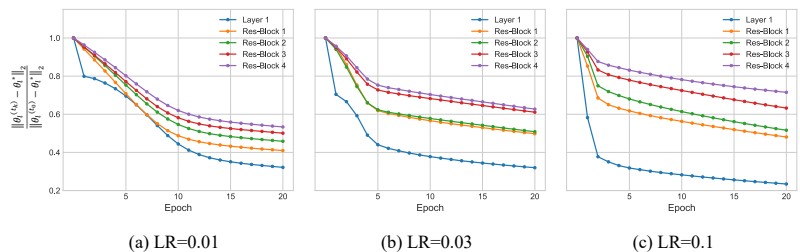

| (a) LR=0.01 | (b) LR=0.03 | (c) LR=0.1 |

Figure 13: The convergence process of FCNNs with different learning rates.

**Weight Decay.** The experiments with FCNNs in previous sections are conducted without weight decay. It is interesting to investigate the sensitivity of the layer-wise model convergence with different weight decay strengths. The results are shown in Fig. 14. We can see that when the weight decay becomes stronger, the residual blocks converge slower in a more and more similar convergence rate. We conjecture the reason is that weight decay dominates the total loss when its coefficient is large. In this way, the layer parameters with similar initialization scales tend to converge in similar speed toward zero. Because the residual blocks have identical architectures, they share the same initial parameter distribution, and converge in the same speed when weight decay is strong.

**Optimizer.** In Section 4, we have discussed the mechanism behind layer convergence bias. The flatter/sharper minimizers of different layers make SGD learn at different speeds. This is because SGD is more good at finding flatter minimizers (Pan et al., 2020). In Fig. 15, we compare SGD with three adaptive optimizers: Adagrad, RMSprop, and Adam. It is evident that with adaptive optimizers, layer convergence bias does not hold anymore. We conjecture the reason behind this

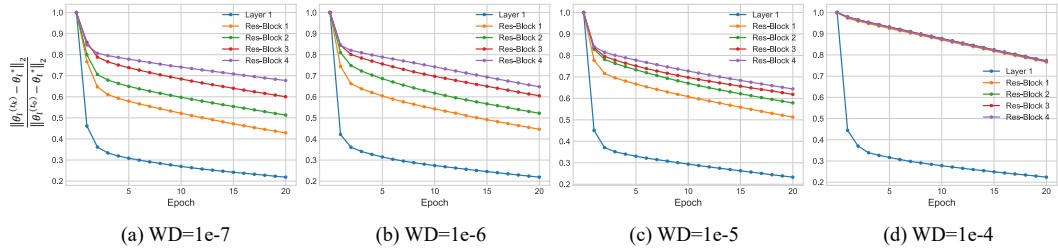

Figure 14: The convergence process of FCNNs with different weight decay strengths.

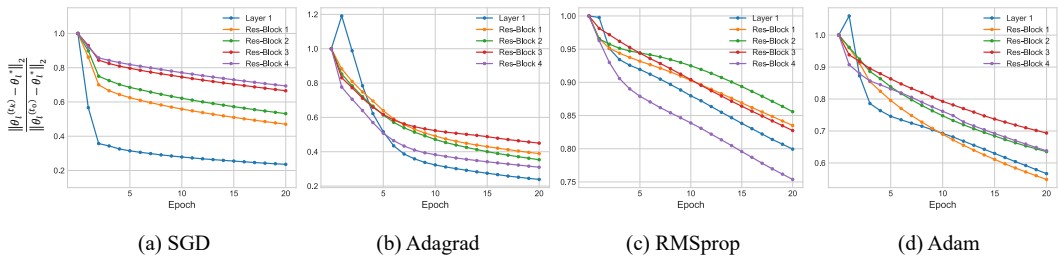

Figure 15: The convergence process of FCNNs with different optimizers.

is that the adaptive optimizers heuristically assign different learning rates for different parameters, making their optimization hardly predictable.

**Normalization Methods.** Like residual connection, batch normalization Ioffe & Szegedy (2015) is also a common design in modern DNN architectures. As discussed in previous literature, normalization in the neural networks helps to make the layer inputs more stable and make the loss landscapes smoother, thus accelerates the model training Santurkar et al. (2018). In Section 3 and Section 4, we mainly use the FCNNs without normalization to verify and explore the layer convergence bias. Here we investigate how the normalization methods (*i.e.*, batch normalization, layer normalization Ba et al. (2016), and group normalization Wu & He (2018)) help the convergence, and whether the shallower layers still converge faster in these cases. As shown in Fig. 16, all layers converge faster when adding batch normalization to them. Particularly, "Res-Block 1" accelerates the most and reach a similar convergence rate as "Layer 1". The layer convergence bias also holds for batch normalization. For layer normalization and group normalization, the models show a significantly faster convergence rate than the model using batch normalization. All layers show effective convergence at an early stage of training (*i.e.*, the first five epochs). In these two cases, different layers have similar convergence rates, thus no evident layer convergence bias emerges.

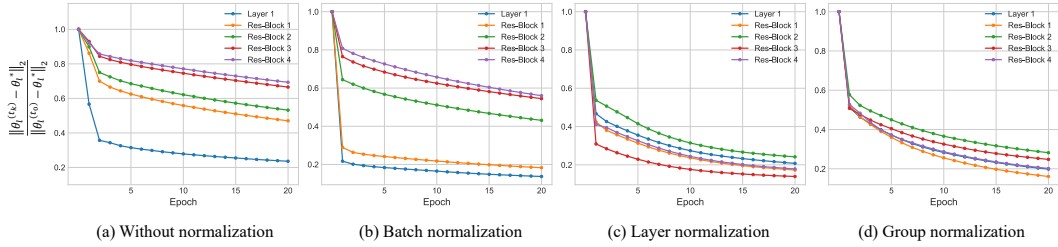

Figure 16: The convergence process of FCNNs with different normalization methods. When using group normalization, we set the group number to 8.

## A.4 RESULTS ON HARDER DATASETS

For verifying the layer convergence bias on more datasets, we show more convergence results on four harder image classification datasets (see Fig. 17). Most of the classes in these datasets only have $< 100$ samples, making them harder to learn. Note that the experiments are conducted with the learning rate of 0.01 (learning rate of 0.1 failed in some cases because these datasets have too many classes but not sufficient samples, leading to non-decreasing loss), some deeper layers have quite similar convergence rates because of the small learning rate. But roughly speaking, layer convergence bias still holds for these datasets.

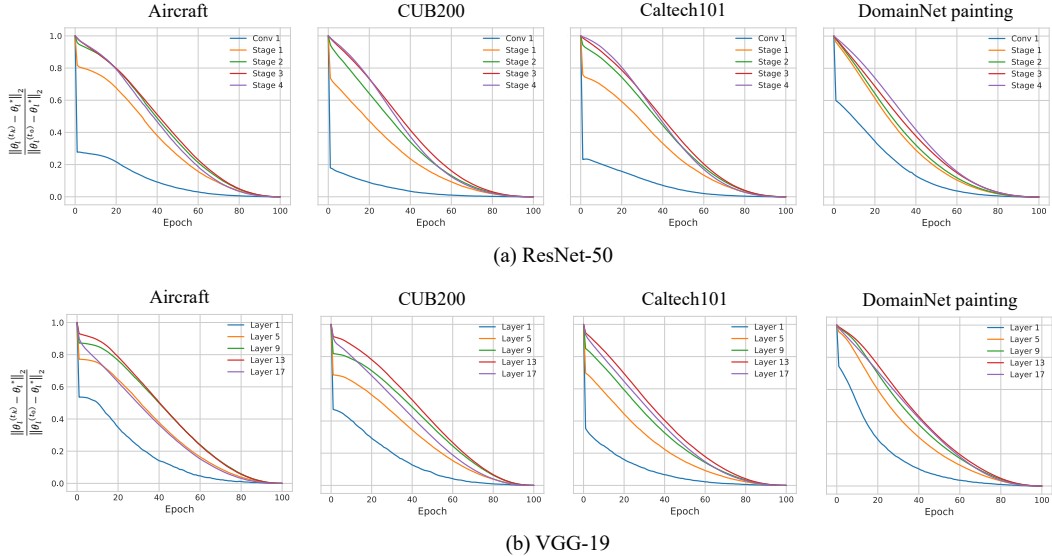

Figure 17: The convergence process of CNNs on four image classification tasks.

## A.5 REPEATABILITY OF THE VISUALIZATIONS

Do different ImageNet trained models produce dramatically different loss landscapes? We plot the loss landscapes of different models with different random seeds in Fig. 18, 19. Quite similar patterns of the landscapes for different layers can be observed on both ResNet and VGG with different random seeds.

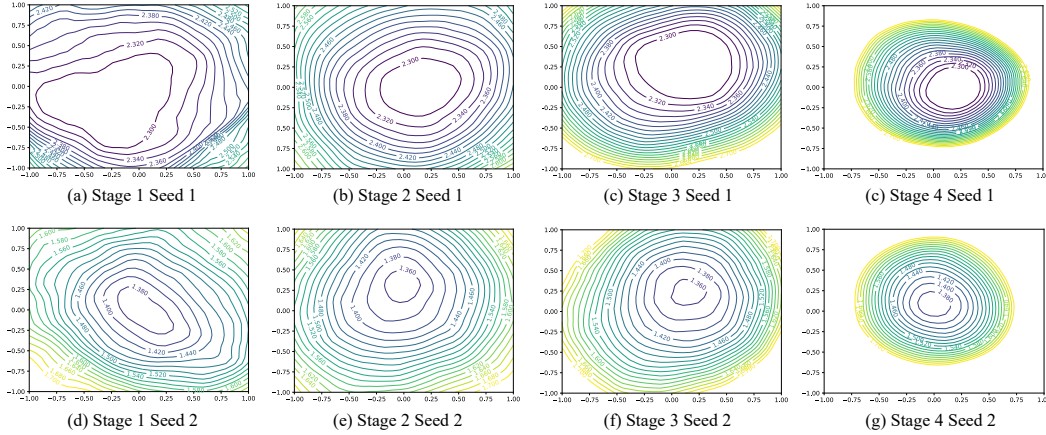

Figure 18: The loss landscapes of different layers of ResNet-50.

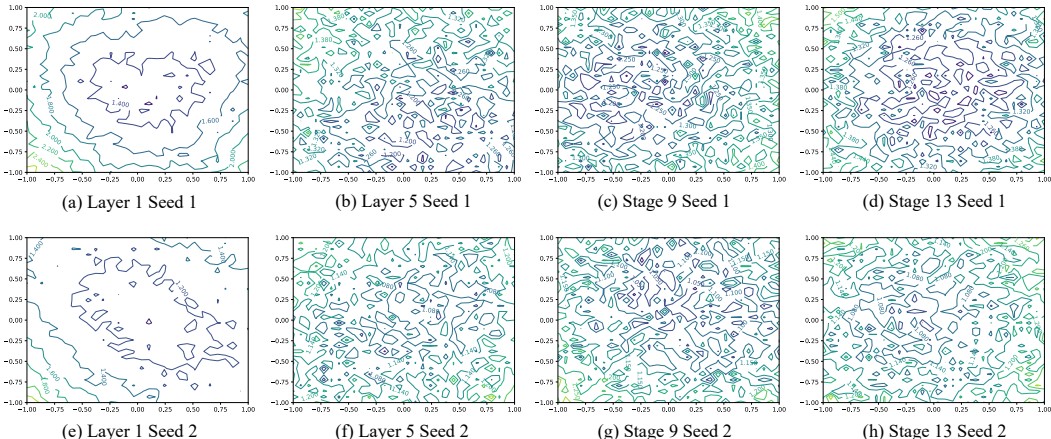

(a) Layer 1 Seed 1     (b) Layer 5 Seed 1     (c) Stage 9 Seed 1     (d) Stage 13 Seed 1

(e) Layer 1 Seed 2     (f) Layer 5 Seed 2     (g) Stage 9 Seed 2     (h) Stage 13 Seed 2

Figure 19: The loss landscapes of different layers of VGG-19.

## A.6 MODELS OBEYING LAYER CONVERGENCE BIAS PERFORM BETTER

In Section 4 and Section 5, it is discussed that layer convergence bias indicates that the shallower layers are learning low-level features (or low-frequency components of the target function). It is reasonable learning low-level features first have greater potential to reach good model performance, since the model can establish its high-level features based on relatively stable low-level feature spaces.

To examine whether the fast establishment of low-level features benefits model performance, we train four different FCNN models with the same amount of parameters, but different architectures, to fit the Sine target with four components. This experiment is based on a finding that a residual block with more layers in it tends to converge more slowly. We construct four FCNN models, each of them has four residual blocks (maybe in different sizes). The convergence processes are shown in Fig. 20. We can see that the blocks with the largest complexity always converge the most slowly. As the block with depth=4 being placed shallower in the FCNN, the regression MSE loss goes higher. In other word, if a shallower layer converge slowly, the model gets poorer performance. This may due to the vulnerability of deeper layers. If they converge based on changing shallower layers, it is hard for them to learn good features based on their unstable inputs.

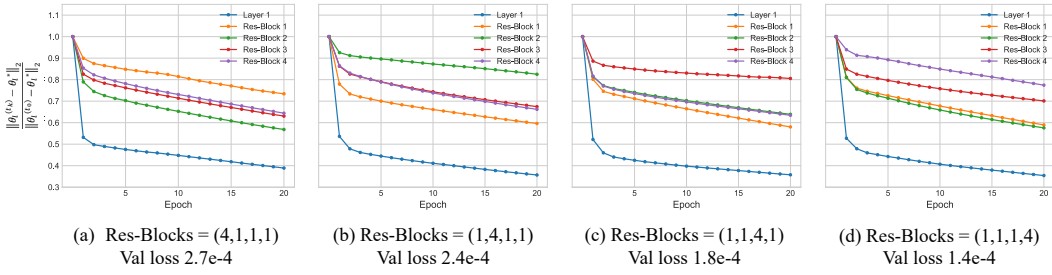

(a) Res-Blocks = (4,1,1,1)
Val loss 2.7e-4

(b) Res-Blocks = (1,4,1,1)
Val loss 2.4e-4

(c) Res-Blocks = (1,1,4,1)
Val loss 1.8e-4

(d) Res-Blocks = (1,1,1,4)
Val loss 1.4e-4

Figure 20: The convergence process of FCNNs with different residual block sizes and their validation performance on the regression task. Each model has a four-layer residual block and three one-layer residual blocks (*e.g.*, "Res-Blocks=(4,1,1,1)" means the first residual block has four layers, and the rest three blocks have only one layer).

The results can also be understood from another perspective. If the deeper block contains more parameters (with more fully connected layers in it), it would be helpful for this block to learn the corresponding high-frequency components of the target function. Therefore, the model can reach better performance. A similar observation is obtained in Section 6.1: when putting wider layers of the ViT deeper, the model can reach higher performance.

## A.7 LAYER CONVERGENCE BIAS FOR VISION TRANSFORMERS

As discussed in Section 6, ViT can benefit from distributing more parameters in the deeper layers. This result comes from one of our main findings about layer convergence bias: the deeper layers tend to learn high-frequency components of the target function, thus converge more slowly. So adding more parameters for the deeper layers is beneficial for these layers to learn the high-frequency components which are usually harder.

When making this claim, we do not verify the layer convergence bias for the ViT. The main difficulty for verifying layer convergence bias for ViTs is brought by its typical training scheme. ViT needs adaptive optimizers to train, otherwise it converges very slowly. However, adaptive optimizers change the learning rates of different parameters according to their optimization procedures. This leads to unfair convergence comparison between layers, thus affects the layer convergence bias, as shown in Fig 15. Therefore, we try both SGD and Adam optimizers for training ViTs on ImageNet, and see whether layer convergence bias holds in some cases. As shown in Fig. 21 (a), the ViT shows a roughly trend of layer convergence bias when optimizing with Adam, where the deepest "Encoder Block 12" converges the slowest. However, some other layers do not strictly obey layer convergence bias (*e.g.*, the shallowest "Patch Embedding" does not learn fastest among all blocks). When optimizing with SGD, the ViT shows a good layer convergence bias. The results indicate that ViTs approximately share the same rules as FCNNs and CNNs, thus supports the discussions in Section 6.

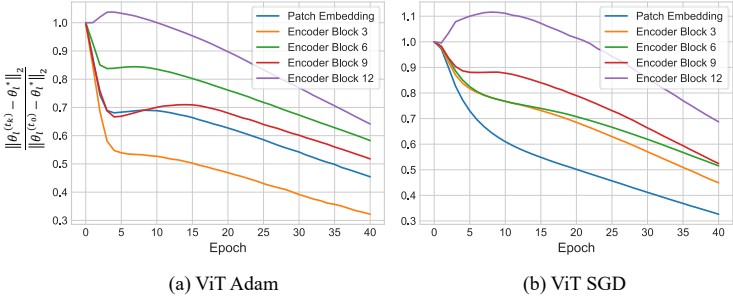

(a) ViT Adam                    (b) ViT SGD

Figure 21: The convergence curves of ViTs on ImageNet with different optimizers. With Adam optimizer, the ViT does not obey the layer convergence bias strictly. While SGD can ensure relatively ideal faster convergence processes of shallower layers.

## A.8 CONNECTION TO LARS OPTIMIZATION SCHEME

One of the most important factors that affect the optimization procedure is the learning rate. In this work, it is shown that the shallower layers can learn effectively with large learning rates, but the deeper layers only learn fast after learning rate decays. Is there any connection between layers and its suitable learning rate?

LARS optimizer You et al. (2017) made a significant contribution to training DNNs with huge batch sizes and large learning rates. The key observation in the literature is that the weight-to-gradient ratio highly varies in different layers. If a layer has greater gradients and relatively smaller weights, it would be hard for it to converge due to the vigorous parameter update. So LARS considers the scale of the weights and its gradient norms in each layer and assigns a local learning rate for a layer to make it converge effectively and stably. For FCNNs in our work, its different hidden layers are initialized with the same scale due to their identical architecture, but the deeper layers usually have larger gradients. As a result, the larger gradients may make these layers struggle to converge. Similarly, the CNNs (*i.e.*, ResNet-50 and VGG-19) have wider deeper layers. These layers have smaller initial parameters, so their gradients may lead to drastic weight variations if the learning rate is too large. In this way, we can understand why they cannot get close to their optimal points effectively at the early stage of training. It explains layer convergence bias from another perspective.

