# OpenReview forum: "Which Layer is Learning Faster? A Systematic Exploration of Layer-wise Convergence Rate for Deep Neural Networks"
_ICLR.cc/2023/Conference — ICLR 2023 poster_

### Official Review · Reviewer_RsgU · 2022-10-24

**Confidence:** 4
**Correctness:** 3
**Technical Novelty And Significance:** 3
**Empirical Novelty And Significance:** 3
**Recommendation:** 8

**Clarity, Quality, Novelty And Reproducibility:**

- **Clarity**: The paper is well-written and easy to follow.
- **Quality**: Some of the arguments of this work could be better supported through experiments that test the causal connections between the different argumentss, and investigated deeper using proper ablation studies with several seeds.
- **Novelty**: As far as I know, this study is novel.
- **Reproducibility**: The lack of studies showing the sensitivity of the observations to different hyperparameters, e.g., learning rate schedule, weight decay, etc, might make some of the observations not fully reproducible in other settings.

**Strength And Weaknesses:**

# Strengths
1. **Interesting empirical study based on an intriguing phenomenon**: The observation that deeper layers have a slower speed of convergence when training on many tasks is interesting and worth investigating. In particular, I find the following observations and experiments noteworthy:
   a. Connection between rate of convergence of deeper layers and frequency of target function.
   b. Consistent observation that deeper layers have a rougher loss landscape.
   c. Faster convergence rates on shallower layers and frequency specialization is not an architectural bias, as it changes during training.
2. **Consistent observations on synthetic and real tasks**: I really appreciate the alternation between experiments on synthetic tasks and real tasks in which the controllable experiments are used to isolate a described effect which can then be observed on the more realistic training runs.
3. **Clear writing**: The paper is clearly written and easy-to-follow. The different sections follow naturally from the previous ones and create a coherent storyline.

# Weaknesses
1. **Lack of causal connection to support arguments in Sec. 6.1**: The explanationss about the usefulness of having wider layers at deeper stages of a neural network are not proprely backed by experiments. In particular, no experiment shows that increasing/decreasing the width of a given layer increases/decreases its speed of convergence. Similarly, the claimed benefits of the proposed architectural changes to ViTs are only superficially investigated. To support these claims, a much further investigation than the one provided should be performed.
2. **Results based on single runs**: Most results are based on observations made for a single run and lack studies of sensitivity to different hyperparameters. In this regard, some of the results presented in this study might be a bit anecdotal.
3. **No normalization of gradients**: While the authors soundly argue that their proposed convergence rate metric needs proper normalization to be comparable among layers, they do not seem to care about normalizing the gradients by number of parameters so that they become comparable among layers. If deeper layers have many more parameters than shallower ones, then it is not surprising that the unnormalized gradient norm of the shallower layers is smaller.
4. (Minor) **Convergence rate metric is not strongly motivated**: The reasons behind using this specific convergence rate metric and not the standard convergence rate used in optimization theory are not very strongly motivated. In particular, the short footnote explaining this on page 2 is a bit unclear.

**Summary Of The Paper:**

This paper presents an empirical study trying to explain the differences in speed of convergence of different layers of a neural network. To that end, the authors propose to measure the speed of convergence by tracking the rate of change in distance to the optimum of different layers within a time span. They argue that, contrary to the gradient magnitude, this metric can capture a more relevant notion of convergence speed consistent with different phenomena. In particular, ther experiments seem to indicate that on many training tasks, deeper layers converge slower than shallower ones, and that they fit higher frequency functional terms. The authors then argue that this inherent difficulty of training deeper layers can explain the success of using wider layers deeper in CNNs. Inspired by this intuition they then suggest and, partially test, using this design choice on modern ViTs.

**Summary Of The Review:**

Overall, I believe this work can be of interest to the community as it provides intuitive insights on an intriguing phenomenon. This being said, I believe these insights coulld be better grounded on certain experiments and isolated with respect to different hyperparameters. If the experiments where more thorough this paper could be a strong contribution. In the current stage, though, I believe this is a borderline paper, slightly above the threshold for acceptance.

----

**Post-rebuttal udpate** : After the detailed rebuttal by the authors where a new improved version of the manuscript has been shared, I have decided to increase my score to an 8, as most of my concerns have been alleviated.

---

> ### Author Response · Authors · 2022-11-17
> **Response to Reviewer RsgU (2/2)**
>
> **Q3**: (No normalization of gradients) While the authors soundly argue that their proposed convergence rate metric needs proper normalization to be comparable among layers, they do not seem to care about normalizing the gradients by number of parameters so that they become comparable among layers. If deeper layers have many more parameters than shallower ones, then it is not surprising that the unnormalized gradient norm of the shallower layers is smaller.
>
> **Response**: Thank you for your valuable comment. We agree that the parameter scales are usually related to the number of parameters in a layer, thus the gradients need to be normalized. In our experiments, we avoided this problem by using an identical structure for all layers that are involved in gradient comparison (as introduced in Sec 3.1). In this way, all layers for comparison had the same initial parameter scale, and their gradients were comparable among layers. As an important perspective, we added the discussion of gradient and parameter scales to Appendix A.8, and how this can inspire us to assign better learning rates to different layers.
>
> ---
>
> **Q4**: (Convergence rate metric is not strongly motivated) The reasons behind using this specific convergence rate metric and not the standard convergence rate used in optimization theory are not very strongly motivated. In particular, the short footnote explaining this on page 2 is a bit unclear.
>
> **Response**: Thank you for your great comment, and sorry for our unclear description in the footnote. The convergence rate proposed in this work focuses on comparing which layer is approaching its optimal point faster at the early training stage. But previous convergence rate in optimization theory is defined as the distance ratio when the optimization step goes to infinity. There are two shortcomings of this convergence rate measurement for comparing the layer-wise convergence. First of all, it is mainly used for measuring the convergence after sufficient iterations, while the layer convergence bias is more significant at earlier stages. Second, the convergence difference between layers is not large enough to be compared at the exponential level. So we proposed to measure the convergence as the distance-to-time ratio. In this way,  the difference can be observed more easily.
>
> To state our motivation in a better way, we removed the footnote and added the description to Sec.2: "It focuses on measuring an exponential level convergence when the optimization step goes to infinity. Since the difference in convergence rates between layers usually appears at an early stage of training, and it is not large enough to compare at an exponential level, we define our new convergence metric to show the convergence difference in a clearer way."

---

> > ### Comment · Reviewer_RsgU · 2022-11-18
> > **Thank you for your reply**
> >
> > Thank you very much for your detailed reply and for improving the manuscript based on the feedback of all the reviewers. Regarding my concerns, most of them have been alleviated. The new ablation studies, as well as the careful discussion about normalization factors and the specific choice of convergence metric make this paper sounder and more thorough. On the other hand, I find the intuitions regarding ViTs to be interesting and valuable, although in my opinion they still are quite preliminary and would probably deserve further study.
> >
> > Overall, I have decided to increase my score to an 8: accept, good paper.

---

> ### Author Response · Authors · 2022-11-17
> **Response to Reviewer RsgU (1/2)**
>
> We would like to express our gratitude to you for your valuable time and effort in reviewing our manuscript and especially for providing constructive comments and suggestions for improving the manuscript. Following your comments and suggestions, we have made changes to the revised manuscript. A point-by-point reply to your comments is given below.
>
> ---
>
> **Q1**: (Lack of causal connection to support arguments in Sec. 6.1) The explanations about the usefulness of having wider layers at deeper stages of a neural network are not properly backed by experiments. In particular, no experiment shows that increasing/decreasing the width of a given layer increases/decreases its speed of convergence. Similarly, the claimed benefits of the proposed architectural changes to ViTs are only superficially investigated. To support these claims, a much further investigation than the one provided should be performed.
>
> **Response**: Thanks for your insightful comments. In Sec. 6.1, the practice of redistributing parameters between layers was motivated by the findings in Sec. 4 and 5, where we found that the deeper layers tend to converge more slowly because they are learning harder task-specific features (or high-frequency components). So adding parameters on the deeper layers may help them to learn the features better. It was an experiment based on intuition, and it worked out as expected. The difficulty in ensuring that increasing/decreasing the parameters of a ViT layer increases/decreases its convergence rate is that ViTs are usually trained with adaptive optimizers like Adam. The parameters thus would be assigned different learning rates, and their behaviors become unstable and less predictable (see Fig. 13 in Appendix A.3). When it comes to SGD, we added an experiment to discuss the causal connection between layer (block) complexity and convergence rate (Appendix A.5). We found that the more complex residual blocks usually converges more slowly. And putting the complex residual blocks deeper in the DNN is beneficial for its fitting performance.
>
> ---
>
> **Q2**: (Results based on single runs) Most results are based on observations made for a single run and lack studies of sensitivity to different hyperparameters. In this regard, some of the results presented in this study might be a bit anecdotal.
>
> **Response**: Thanks for your constructive comment. To ensure the findings in this work are stable, and to investigate their sensitivity to different hyperparameters, we added experiments with different training schemes to Appendix A.3. From the reported results, we can see that: 1) layer convergence bias holds for models with different depths; 2) with lower learning rate (e.g., from 0.1 to 0.01), the layer convergence bias gradually becomes weaker because the gradient predictiveness w.r.t. parameters of all layers get close to 1, thus do less harm to the convergence rate of deeper layers; 3) with strong weight decay, the layer parameters initialized in similar scales converge similarly because they learn towards zero in a similar speed; 4) adaptive optimizers usually doesn't show layer convergence bias because they can assign different learning rates for different parameters according to their optimization behaviors; 5) batch/layer/group normalization accelerates the convergence of some layers, and sometimes breaks layer convergence bias.

---

### Official Review · Reviewer_xk3R · 2022-10-26

**Confidence:** 3
**Correctness:** 3
**Technical Novelty And Significance:** 3
**Empirical Novelty And Significance:** 4
**Recommendation:** 6

**Clarity, Quality, Novelty And Reproducibility:**

Clarity
----------
The paper is overall clear, describing in an understandable way the phenomenon observed and the investigative steps taken.

A few potential improvements:
1. The convergence rate in Def 2.1 does not seem to correspond to the convergence curves in most of the following figures.
2. The difference between "first layer" and "first hidden layer" in Sec 3.1 is confusing, usually when a difference is made, that's because the "first layer" is the input one, which does not have any parameters.
3. Figure clarity could benefit from having the setup explicitly stated in the caption (which dataset, which experiment is being performed...), as well as the conclusions and points of interest (e.g., what's happening at Epoch 150 in Fig. 3?)
4. In Def 4.1, is it intended to have $G''_{l,t}$ defined with $\theta^{(t+1)}_L$? Why is $L$ updated, in addition to $1, \ldots, l-1$?


Quality
-----------
The experiments are well designed, and are consistent with the conclusions made. The experiments on frequency (in Sec. 5) and transfer learning (Sec. 6.2) are interesting and insightful. But I'm not sure they support it to the extent being claimed.

1. I don't really agree with naming the investigated phenomenon "bias", although it shares similarities with the "spectral bias" (and seems related to it), it is fundamentally different as it does not directly affect the behavior or predictive power of the network at any given time, it's just related to internal training dynamics. I don't think we can say that the network exhibits a bias (in its abilities, or performance, or predictions) due to this phenomenon.
2. The paper assumes that weights of each layer are converging to a value, but it does not report whether it actually does before training stops (at epoch 100 or so depending on the experiment).
3. The observations are made on typical architectures, known for working well experimentally. It is not clear they are intrinsic to neural network training, correlated to good models and training conditions, or if there is any causality link: are there models where the output layers converge first, but that don't perform well? Fig. 8 (e-h) shows that changing the input distribution can change the convergence behavior. Section 6.1 might be affecting the convergence behavior by changing the architecture, but it's not clear if it actually changes, or this is what causes the performance improvements.
4. The visualization of loss landscape "slices" is interesting, and consistent with the idea of smoother, wider minima in the lower layers, but I don't think it's a definite proof, let alone a causal explanation. I don't think that this being "the fundamental reason behind this phenomenon" is being adequately supported.
5. It would be good for figures 1(b), 8, and 11-13 to show the convergence all the way to $\theta_l^*$, in addition to the zoomed-in portion at the beginning.


Novelty
-----------
The paper clearly states that faster convergence in the first layers has been observed and reported in cited work earlier. I think it would have been better to state more prominently towards the beginning of the paper that it builds on these observations, rather than in the last paragraph before the conclusion.

It is good to have an explicit consideration for skip-connections, given that they're often used in modern deep architectures, but it would have been nice to also investigate batch normalization or layer normalization, at least in the context of ResNets, to see if the findings still hold.

Reproducibility
--------------------
The paper, formulas, and description of experiments are defined clearly enough that someone with knowledge of the field could reimplement and reproduce the results.
However, the source code is not provided with the paper.

**Strength And Weaknesses:**

Strengths
-------------
1. Confirms observations of faster convergence in the first layers during training of a DNN, using a quantitative metric
2. Establishes a link between learning lower frequencies earlier, and learning them in the first layers, showing that higher layers converge faster if they don't have any high-frequency components to learn
3. Empirically explores loss landscapes of different layers, for different architectures (FCNN, VGG, ResNets)
4. Exploits these observations to propose a better repartition of parameters for vision transformers

Weaknesses
-----------------
1. I don't think the reason behind these observations have been convincingly proven. The explanations provided are consistent, but may not be the only ones.
2. Previous work should be more prominently mentioned, as these observations are not new.

**Summary Of The Paper:**

The paper investigates the observation that, during the training of deep neural networks, lower layers ("shallower", closer to the input) converge faster towards their final value, compared to upper layers ("deeper", further away from the input).
It quantifies that tendency on several architectures, on synthetic data and natural images.
It then draws links with the "spectral bias" of neural networks, showing that lower layers seem to learn functions with lower frequencies (in the input space), and higher layers learn, more slowly, the higher-frequency components (if any).

**Summary Of The Review:**

This is a really interesting exploratory work, and the explanations advanced are plausible, but not really convincing. It is not clear how much of these findings can generalize to other architectures, or distributions other than natural images and synthetic 1D data.
I'm leaning towards accepting it.

**Update after responses**
Thanks for your answers and clarifications.
While I recognize they add insight, I still think the paper is borderline, and I'm keeping my rating.

---

> ### Author Response · Authors · 2022-11-17
> **Response to Reviewer xk3R (3/3)**
>
> **Q8**: The observations are made on typical architectures, known for working well experimentally. It is not clear they are intrinsic to neural network training, correlated to good models and training conditions, or if there is any causality link: are there models where the output layers converge first, but that don't perform well? Fig. 8 (e-h) shows that changing the input distribution can change the convergence behavior. Section 6.1 might be affecting the convergence behavior by changing the architecture, but it's not clear if it actually changes, or this is what causes the performance improvements.
>
> **Response**: Thank you for your insightful comments. To show the causality link between the model performance and the layer-wise convergence procedure, we designed an experiment to deliberately make some shallower layers converge more slowly than the deeper layers. The experiment and its results are added to Appendix A.5.  From the results, we know that with the same number of model parameters but different model architectures, the model that learns with a faster shallower-layer convergence rate has a higher performance than the models whose shallower layers learn more slowly. The results may show that it is beneficial for the models to learn low-level features first, and gradually refine the high-level features.
>
> ---
>
> **Q9**: The visualization of loss landscape "slices" is interesting, and consistent with the idea of smoother, wider minima in the lower layers, but I don't think it's a definite proof, let alone a causal explanation. I don't think that this being "the fundamental reason behind this phenomenon" is being adequately supported.
>
> **Response**: Thank you for your comment. We agree with you that it isn't definite proof. This work provides an intuitive explanation for the "layer convergence bias" phenomenon by examining the loss landscapes of different layers. For rigorously proving its theoretical mechanism, we think mathematical derivations from the perspective of data distribution, feature, and gradient variation through layers may be promising for future research.
>
> ---
>
> **Q10**:  It would be good for figures 1(b), 8, and 11-13 to show the convergence all the way to θl*, in addition to the zoomed-in portion at the beginning.
>
> **Response**: Thank you for your great suggestion. We've tried combining the complete convergence curves and a zoomed-in portion to a single figure but found it became too complex, and it is difficult for readers to find the important difference between layers quickly.
>
> ---
>
> **Q11**: The paper clearly states that faster convergence in the first layers has been observed and reported in cited work earlier. I think it would have been better to state more prominently toward the beginning of the paper that it builds on these observations, rather than in the last paragraph before the conclusion.
>
> **Response**: Thanks for your suggestion. We agree that stating it earlier in the paper is a good idea. But our findings are not built directly on the observations of [2]. This literature showed that the learned features of a CNN seemed to stabilize earlier in the shallower layers than in the deeper layers. However, our findings were obtained from the perspective of parameter convergence. So [2] can be a good support for our explanation of layer convergence bias, especially on CNNs. Following your suggestion, we removed the description in Sec. 7 and added the sentence to Sec. 4: "Before our work, (Zeiler & Fergus, 2014) also revealed that the general features in a five-layer CNN stabilized faster than the specific features."
>
> [2] Zeiler M D, Fergus R. Visualizing and understanding convolutional networks[C]//European conference on computer vision. Springer, Cham, 2014: 818-833.
>
> ---
>
> **Q12**: It is good to have an explicit consideration for skip-connections, given that they're often used in modern deep architectures, but it would have been nice to also investigate batch normalization or layer normalization, at least in the context of ResNets, to see if the findings still hold.
>
> **Response**: Thank you for your insightful suggestion. Batch normalization has been proven to make the training converge faster because it can make the loss landscapes smoother [3]. Similarly, our work demonstrates that deeper layers tend to have more rugged loss landscapes than the shallower layers. So we added experiments to Appendix A.3 to investigate how batch/layer/group normalization affect FCNNs with residual connections.
>
> [3]Santurkar S, Tsipras D, Ilyas A, et al. How does batch normalization help optimization?[J]. Advances in neural information processing systems, 2018, 31.

---

> ### Author Response · Authors · 2022-11-17
> **Response to Reviewer xk3R (2/3)**
>
> **Q4**: Figure clarity could benefit from having the setup explicitly stated in the caption (which dataset, which experiment is being performed...), as well as the conclusions and points of interest (e.g., what's happening at Epoch 150 in Fig. 3?)
>
> **Response**: Thank you for your nice suggestion. We actually missed some important setups in the captions, which may probably confuse the readers. As suggested, we revised the caption of Fig 3 to: "The gradient predictiveness of shallower and deeper layers of FCNN. The learning rate decreases from 0.1 to 0.01 at Epoch 150." We also revised the caption of Fig. 5 to: "The loss landscapes of different layers of ResNet-50 (a,b) and VGG-19 (c,d) on ImageNet." And the caption of Fig. 6 was revised to: "The visualization of what each residual block of the FCNN learns."
>
> ---
>
> **Q5**: In Def 4.1, is it intended to have $G''_{l,t}$ defined with $\theta_L^{(t+1)}$? Why is L updated, in addition to 1,…,l-1?
>
> **Response**: Thanks for your kind reminder, and sorry for the incorrect definition of Gl,t''. Actually, in this definition, all layers but layer l are updated. So it should be the layers 1, …, l-1, and l+1, …, L are updated. But we mistakenly wrote $\theta_{l+1}^{(t+1)}$ to $\theta_{l+1}^{(t)}$. Therefore, we revised the $\theta_{l+1}^{(t)}$ to $\theta_{l+1}^{(t+1)}$.
>
> ---
>
> **Q6**: I don't really agree with naming the investigated phenomenon "bias", although it shares similarities with the "spectral bias" (and seems related to it), it is fundamentally different as it does not directly affect the behavior or predictive power of the network at any given time, it's just related to internal training dynamics. I don't think we can say that the network exhibits a bias (in its abilities, or performance, or predictions) due to this phenomenon.
>
> **Response**: Thank you for your constructive comment. We were actually inspired by "spectral bias" to name our main finding in this work. In that literature [1], the author defined the phenomenon: "we find that these networks prioritize learning the low-frequency modes, a phenomenon we call the spectral bias." Similarly, the layer convergence bias depicts a phenomenon that the shallower layers tend to converge faster, because the low-level basic features are easier for these layers to learn. It does not only related to internal training dynamics but also exhibits a bias related to its low-level and high-level (semantics) abilities.
>
> [1] Rahaman N, Baratin A, Arpit D, et al. On the spectral bias of neural networks[C]//International Conference on Machine Learning. PMLR, 2019: 5301-5310.
>
> ---
>
> **Q7**: The paper assumes that weights of each layer are converging to a value, but it does not report whether it actually does before training stops (at epoch 100 or so depending on the experiment).
>
> **Response**: Thank you for your valuable comment. In this work, we ensured the convergence of parameters from two aspects: 1) Long training. Most models were trained for 200-300 epochs to ensure that they had learned the task well. For ImageNet classification, we trained the CNNs for 120 epochs, which is 600,000 iterations. All the convergence curves were ensured to be well-behaved. 2) Learning rate decay. As the learning rates decay to small values, the model parameters finally get stable near a local optimum. So it can be safely concluded that the models are converged.

---

> ### Author Response · Authors · 2022-11-17
> **Response to Reviewer xk3R (1/3)**
>
> We would like to express our gratitude to you for your valuable time and effort in reviewing our manuscript and especially for providing constructive comments and suggestions for improving the manuscript. Following your comments and suggestions, we have made changes to the revised manuscript. A point-by-point reply to your comments is given below.
>
> ---
>
> **Q1**: I don't think the reason behind these observations have been convincingly proven. The explanations provided are consistent, but may not be the only ones.
>
> **Response**: Thank you for your valuable comment. There may be multiple reasons why different layers converge in different speeds. For example, different layers may be initialized in different scales, a consistent learning rate makes the layers with smaller parameters harder to learn effectively due to the large update steps. Related discussions about the scale of gradients and parameters were added to Appendix A.8.
>
> ---
>
> **Q2**: The convergence rate in Def 2.1 does not seem to correspond to the convergence curves in most of the following figures.
>
> **Response**: Thank you for your constructive comment. In most of the convergence curves, we showed how the distances between different layers' current parameters and their convergence points change over time. The convergence rate defined in Def 2.1 is the distance-to-time ratio, which can be observed as the slopes of the convergence curves. The reason why we draw convergence curves like this is that the scales of all layers are consistently [0,1], and it would be convenient for observing how much the parameters are approaching their optimal points, and comparing the convergence between layers.
>
> ---
>
> **Q3**: The difference between "first layer" and "first hidden layer" in Sec 3.1 is confusing, usually when a difference is made, that's because the "first layer" is the input one, which does not have any parameters.
>
> **Response**: Sorry for the unclear description. In this work, the "first layer" and "first hidden layer" are two different layers. For the FCNN with the structure (1-32-32-32-1), there are four fully connected layers, "first layer" and "first hidden layer" are the (1-32) layer and the first (32-32) layer, respectively. They refer to the fully connected layers instead of the neurons. To avoid ambiguity, we revised the description in Sec 3.1 to "In the following analysis, the first fully-connected layer (1-32) is named Layer 1, and the subsequent two layers (32-32) are called Hidden layer 1, Hidden layer 2 respectively."

---

### Official Review · Reviewer_w2MX · 2022-10-29

**Confidence:** 3
**Correctness:** 3
**Technical Novelty And Significance:** 2
**Empirical Novelty And Significance:** 3
**Recommendation:** 6

**Clarity, Quality, Novelty And Reproducibility:**

The manuscript is well-written and communicates the core principles and hypotheses clearly. The authors use pedagogical examples of simplified target functions to establish the differences in learning dynamics across layers of a neural network, with observations that are consistent with empirical evidence in the existing literature. The work provides sufficient information to replicate core results from a reproducibility standpoint. Furthermore, the paper presents sufficiently novel results, particularly the notion of gradient predictability that could provide useful diagnostic tools to debug learnability in future architecture designs.

**Strength And Weaknesses:**

Some strengths of the work include the following:
+ The paper is clearly written, with well-grounded motivations and pedagogical examples on understanding the learning dynamics across different layers of canonical neural network architectures.
+ The authors highlight the limitations of gradient magnitude as a measure of learning progress, instead proposing a proxy that measures the predictability of gradients as learning progresses.
+ The insights are leveraged to provide practical design guidance and explain known phenomena in the literature (transfer learning accelerates training on downstream tasks).

Some weaknesses of the current draft are outlined below:
- The critical observations of the work, i.e., layers converge at different rates and learn different components of the target functions, are well established in the literature. However, it would be helpful to clarify the utility/novelty of the discussion using simplified target functions (Section 3).

**Summary Of The Paper:**

Training neural networks on large-scale real-world datasets typically involve gradient-based updates of deep networks. Therefore, understanding the learning dynamics of these models is critical from the perspective of interpretability, as well as designing better architectures/training algorithms. In this work, the authors study the convergence of different layers in a model across training and demonstrate that early layers in canonical deep learning architectures converge faster and learn low-frequency components of a target function. A central claim of the hypothesis is that gradient magnitude is insufficient to characterize the learnability across layers and instead proposes the notion of gradient predictability as a proxy for estimating learning progress. With analysis across simplified target functions and realistic datasets/models (CIFAR/Resnets), the authors show that their notion of measuring learning progress is consistent with existing work in the literature. Further, the insights are used to motivate and explain design choices in the context of transformer architecture and the benefits of transfer learning.

**Summary Of The Review:**

Summarily, the authors consider the learning dynamics of canonical deep learning architectures through the lens of gradient predictability across network layers. Using simplified target functions (mixture of sinusoids at different frequencies), the authors establish earlier network layers tend to learn lower-frequency components (and converge faster). In contrast, the last layers learn high-frequency components. Experiments across multiple model families and datasets suggest the ubiquity of this finding in convolutional architectures. Finally, these insights are used to guide design choices for better architecture design in the context of vision transformers and explain the efficacy of transfer learning in accelerating learning on downstream tasks.

---

> ### Author Response · Authors · 2022-11-17
> **Response to Reviewer w2MX**
>
> We would like to express our gratitude to you for your valuable time and effort in reviewing our manuscript and especially for providing constructive comments and suggestions for improving the manuscript. The reply to your comments is given below.
>
> ---
>
> **Q1**: The critical observations of the work, i.e., layers converge at different rates and learn different components of the target functions, are well established in the literature. However, it would be helpful to clarify the utility/novelty of the discussion using simplified target functions (Section 3).
>
> **Response**: Thank you for your great suggestion. In Section 3, we used FCNNs to fit the target function with four sine components to show that, shallower layers in a DNN usually converge faster than the deeper layers. This is our main novelty in this work, namely layer convergence bias. We also used simplified targets with 1/2/3 sine components in Section 4 to show whether layer convergence bias also happens for simplified targets. Through experiments and discussions, we showed that with simpler target functions, the convergence of higher layers would be evidently faster than its convergence speed on complex target functions (Fig 8 (a)-(d)). This result also helps to show that the deeper layers have some connections with the higher-frequency components of the target function.

---

### Official Review · Reviewer_ogf3 · 2022-11-01

**Confidence:** 3
**Correctness:** 3
**Technical Novelty And Significance:** 2
**Empirical Novelty And Significance:** 3
**Recommendation:** 5

**Clarity, Quality, Novelty And Reproducibility:**

-The paper is overall clear however some parts do not emphasize the takeaway message making it challenging to follow some of the authors more subtle observations (e.g. 6.2). Some key figure's such as 1 is are bit small hurting readability.

-The work is original however some of the ideas have been described in related work

-The paper seems possible to reproducible

**Strength And Weaknesses:**

*Strengths*

-The observations are intuitive and may help provide insights on the training dynamics of NNs

-To the best of the reviewers knowledge the layer convergence bias has not been pointed out in the literature



*Weakness*

-Article uses a specific definition of convergence that normalizes across layers. Although it seems sensible this is the basis of the entire article so it would be good to fully justify it and discuss other potential notions of comparing convergence across different sets of parameters. Are there limitations to this notion of convergence?

-Some of the conclusions are made on rather limited examples
   a)The main results are shown for a single task and with convolutional networks, how does the architecture affect these conclusions? Would MLP or ViT have similar results?
   b) Gradient “predictiveness”  is concluded to be higher based on 2 layers of one specific model in one specific training scenario analyzed. -The difference between the layers does not really look significant on the shown figures especially with the noise in the measurements
-The novelty to some existing work is not strictly made clear. For example the authors note the observations of the classic Zeiler et al.
-The practical applications for DNN architecture design is not completely convincing. First of all the authors have not even confirmed the results on ViT. Secondly the connection to the observations and making the deeper layers wider is a bit unclear.
- It was not clear what the practical observations of 6.2 were
- It seems a more natural application of such results would be in the realm of optimization, driving new layerwise adaptive optimization schemes. It would be interesting if the authors can also comment on the connection of their work to the popular LARS optimization scheme (You et al "Large Batch Training of Convolutiona Networks")


**Summary Of The Paper:**

The authors study a specific notion of convergence and compare the convergence of different layers. It is found that gradient norm does not correlate to the convergence, early layers converge faster and learn low frequency components

**Summary Of The Review:**

The paper proposes some original observations regarding layerwise convergence. The paper is built on a specific definition which is not fully analyzed. Additionally the experimental results focus on CNN and image data, while the claims suggest it is more general. The practical applications of the results as shown by the authors are not completely clear.

---

> ### Author Response · Authors · 2022-11-17
> **Response to Reviewer ogf3 (2/2)**
>
> **Q4**: The practical applications for DNN architecture design is not completely convincing. First of all the authors have not even confirmed the results on ViT. Secondly the connection to the observations and making the deeper layers wider is a bit unclear.
>
> **Response**: Thank you for your constructive comments. ViTs with three different architecture settings achieved performance of 80.75%, 78.88%, and 75.75% on ImageNet. Because the deeper layers are modeling high-frequency components of the target function, they need a larger model capacity to fit well than the shallower layers. Our result shows that putting more parameters in the deeper layers (making it wider) is helpful in increasing the performance of the ViT. As we know, most transformers have identical encoders throughout all layers, so this finding may lead to a potential design for more powerful Transformers.
>
> To confirm the results, we added the performance of three ViTs to Sec 6.1. In addition, to better support the connection between performance and layer complexity distribution, we added the related experiments and discussions to Appendix A.5. We used the FCNN as a representative model and found that when complex layers are put deeper into the model, it can also reach a better performance after convergence.
>
> ---
>
> **Q5**: It was not clear what the practical observations of 6.2 were.
>
> **Response**: Thanks for your comment. Section 6.2 provides a new perspective on explaining the convergence acceleration effect of transfer learning. We attribute the main reason for faster training with pre-trained parameters to 1) the pre-trained shallower layers are nearly optimal, so they don't present fast convergence in transfer learning; 2) although the pre-trained deeper layers are not as optimal as the shallower layers do, their loss landscapes are much flatter than training from scratch, which makes them converge much faster. We revised the text in Sec 6.2 to clarify these two points.
>
> ---
>
> **Q6**: It seems a more natural application of such results would be in the realm of optimization, driving new layerwise adaptive optimization schemes. It would be interesting if the authors can also comment on the connection of their work to the popular LARS optimization scheme (You et al "Large Batch Training of Convolutional Networks").
>
> **Response**: Thanks for your constructive suggestion. LARS does have some interesting connections with the observations in this work. It made a significant contribution to training DNNs with huge batch sizes and large learning rates. The key observation in the LARS optimization scheme is that the weight-to-gradient ratio highly varies in different layers. It would be hard to converge if a layer has larger gradients and relatively smaller weights. So it considers the scale of the weights and its gradient norms in each layer, and assigns a local learning rate for a layer to make it converge effectively and stably. For DNNs in our work, the slow convergence of deeper layers may also be correlated to their large gradients or small parameter initializations. The citation was added to Sec. 7, and a more elaborated discussion was added to Appendix A.8.

---

> ### Author Response · Authors · 2022-11-17
> **Response to Reviewer ogf3 (1/2)**
>
> We would like to express our gratitude to you for your valuable time and effort in reviewing our manuscript and especially for providing constructive comments and suggestions for improving the manuscript. Following your comments and suggestions, we have made changes to the revised manuscript. A point-by-point reply to your comments is given below.
>
> -----
>
> **Q1**: Article uses a specific definition of convergence that normalizes across layers. Although it seems sensible this is the basis of the entire article so it would be good to fully justify it and discuss other potential notions of comparing convergence across different sets of parameters. Are there limitations to this notion of convergence?
>
> **Response**: Thank you for your great suggestions. This work comes up with a very straightforward convergence measurement (normalized distance to time ratio) to compare convergence across different layers. As another potential notion, we can regard the normalized weight variation (weight variation in an epoch / the weight norm) as an indicator of parameter convergence. Related experimental results and discussions were added to Appendix A.2. The results show that this convergence metric does not work well for comparing the convergence rate between layers because it neglects the learning directions.
>
> As for the limitation of our proposed notion of convergence, we think it somewhat depends on stable convergence. Without a stable convergence point (DNNs may not converge well in some tasks, e.g., reinforcement learning), the measurement doesn't work anymore because for calculating the convergence rate, we need the distance between the parameters and final parameters. To overcome this problem, we trained the DNNs for enough epochs.
>
> -----
>
> **Q2**: Some of the conclusions are made on rather limited examples a)The main results are shown for a single task and with convolutional networks, how does the architecture affect these conclusions? Would MLP or ViT have similar results? b) Gradient “predictiveness” is concluded to be higher based on 2 layers of one specific model in one specific training scenario analyzed. The difference between the layers does not really look significant on the shown figures especially with the noise in the measurements.
>
> **Response**: Thank you for your comments. a) In this work, we used both MLP and CNNs to show layer convergence bias. However, to avoid ambiguity, we named MLPs as FCNNs (fully-connected neural networks) because they sometimes contain residual connections. For ViT's layer-wise convergence processes, we added them to Appendix A.7. The results show that ViT also presents a similar layer convergence bias.  b) For the loss landscapes, we visualized them for both FCNNs and CNNs to ensure that all general cases are covered. The function of "gradient predictiveness" was to introduce our motivation to explore the loss landscapes of DNNs. We surprisingly found that the subtle difference of "gradient predictiveness" between layers can lead to significant convergence behaviors, and therefore dived into the loss landscapes to investigate the reason behind this.
>
> -----
>
> **Q3**: The novelty to some existing work is not strictly made clear. For example the authors note the observations of the classic Zeiler et al.
>
> **Response**: Thanks for your insightful comment. As far as we know, the only work that has a similar observation to ours is Zeiler et al [1]. Unlike this literature, which only discussed the feature variation of different layers of CNNs, our work systematically verifies the layer convergence bias for various DNN architectures. In addition, we found potential explanations for this phenomenon by illustrating the loss landscape curvatures of different layers. In this way, why deeper layers converge more slowly can be understood straightforwardly. Furthermore, we combined our findings with a recently developed theory, spectral bias, to reveal the connection between layers and the feature difficulty of the targets they are modeling.
>
> [1] Zeiler M D, Fergus R. Visualizing and understanding convolutional networks[C]//European conference on computer vision. Springer, Cham, 2014: 818-833.

---

### Author Response · Authors · 2022-11-17
**Summary of Changes of New Version**

We would like to thank the reviewers for their thorough feedback. In response to the many valuable suggestions and questions they provided, we have made substantial revisions to the paper. Please note that we have marked the revisions in blue in the new manuscript. In this comment, we summarize those changes section by section.

-----

**Changes throughout the paper**:

As suggested by Reviewer xk3R, we revised the caption of Fig. 3, 5, and 6 to have the model setup explicitly stated in the caption.

-----

**Abstract and Section 1**: Unchanged.

-----

**Section 2**: The footnote was removed to avoid ambiguity. Instead, we added the discussion of the typical definition of convergence rate and the motivation of our convergence metric to the main text.

-----

**Section 3**:

* It is a bit confusing for "Layer 1" and "Hidden layer 1", so we emphasized that they refer to the fully connected layers instead of the neurons to avoid ambiguity.
* We added the description that our convergence metric is crucial for observing the layer convergence bias. The link to the related discussion in the appendix was also added.
* We added the link to the appendix for factors that would affect the phenomenon.

-----

**Section 4**:

* The incorrect $\theta_{l+1}^{(t)}$ was revised to $\theta_{l+1}^{(t+1)}$ in the definition of $G''_{l,t}$.
* We added the description of a similar observation reported in [1] to support our explanation of layer convergence bias for CNNs.

[1] Matthew D Zeiler and Rob Fergus. Visualizing and understanding convolutional networks. In European conference on computer vision, pp. 818–833. Springer, 2014.

-----

**Section 5**: Unchanged

-----

**Section 6**:

* We confirmed the result of ViTs with different architectures and added the link to the appendix for 1) the causal connection between layer complexity and model performance and 2) layer convergence bias for ViTs.
* We clarified the findings of transfer learning experiments. The analyses of why layer convergence bias does not hold for pre-trained models were added.

-----

**Section 7**:  We added the reference to LARS and the link to the appendix subsection for discussing how the observations in LARS are also helpful in explaining layer convergence bias.

-----

**Appendix**:

* We added the subsection "Convergence Measurement Using Weight Variation" to introduce another potential convergence metric, and showed how the direction information in our previous convergence metric is crucial for it to show layer convergence bias.
* We added the discussion of weight decay and normalization methods to the subsection "Factors Affecting Layer Convergence Bias". They can complement the results and help to show how our results are sensitive to hyperparameters.
* We added the subsection "Models Obeying Layer Convergence Bias Perform Better" to build the causal connection between the layer convergence rate and the model performance.
* We added the subsection "Layer Convergence Bias for Vision Transformers" to show whether layer convergence bias holds for ViTs with different optimizers.
* We added the subsection "Connection to LARS Optimization Scheme" to discuss an insightful observation of layer gradient and weight scales, and how it is related to our findings.

---

### Public Comment · ~Linara_Adilova1 · 2023-05-03
**Some questions**

Hello! Congratulations with the paper. I was trying to reach you at the ICLR chat, but I guess it does not send any notifications, so I ask my questions here.

I was wondering if you define the optimal state for the convergence rate just as the final state achieved during training?

Also - how exactly do you visualize the separate layers? It seems a bit controversial to me, that shallow layers are flat and therefore converge faster. First, being flat does not guarantee faster convergence - with small learning rate it actually can take even longer. Second, overall shallow layers are more sensitive to changes than deeper layers (just intuitively and also for example from https://arxiv.org/abs/1902.01996) so it is strange to see visualization that shows opposite.

---

> ### Author Response · Authors · 2023-05-04
> **Response to Linara Adilova**
>
> Hi!
>
> In this paper, we simply used the final parameters as the optima for each layer, given two mild conditions: 1) the training is long enough; 2) the final learning rate is small enough to make the parameters captured by the local optima.
>
> Regarding your question about our visualization of the layer-wise loss landscapes, we agree with you that a flat landscape doesn't guarantee faster convergence. In the paper, we always begin a relatively large LR (e.g., 0.1 for SGD), which is a common practice and can provide better generalization (**Towards Explaining the Regularization Effect of Initial Large Learning Rate in Training Neural Networks**, NeurIPS 2019, Yuanzhi Li et al.) Actually, if we use very small LRs, the parameters of the deeper layers would easily be stuck in nearby sharp local minima, and the shallower layers need a very long time to converge instead.
>
> Finally, the sensitivity of the layer's reinitialization during training is consistent with our conclusions. If a layer's parameters converge fast (i.e., shallower layers), the reinitialization would be fatal to the learned knowledge, confirming its sensitivity.

---

> > ### Public Comment · ~Linara_Adilova1 · 2023-05-11
> > **Reply**
> >
> > Thanks for the reply!
> >
> > > Actually, if we use very small LRs, the parameters of the deeper layers would easily be stuck in nearby sharp local minima, and the shallower layers need a very long time to converge instead.
> >
> > Did you observe this in your experiments?
> >
> > > Finally, the sensitivity of the layer's reinitialization during training is consistent with our conclusions. If a layer's parameters converge fast (i.e., shallower layers), the reinitialization would be fatal to the learned knowledge, confirming its sensitivity.
> >
> > Sure, but is not it inconsistent with flatness? Flatness should mean that the changes in the parameters do not change the loss - which is opposite to sensitivity/fast convergence.

---

> > > ### Author Response · Authors · 2023-05-13
> > > **Response 2**
> > >
> > > We didn't try very small LRs in our exploration since it may take a long time. This is just a reasonable guess. And we have to apologize for missing one point. The loss landscapes discussed in the paper were the landscapes near the optima instead of the whole picture. And the converging process and the sharpness in different layers may vary with different training strategies. As discussed in the paper, with smaller LRs, different layers may converge with a more similar rate due to their gradient predictiveness w.r.t. parameters may be closer (see Section 4 and Fig 13. in the appendix) instead of with the deeper layers faster.
> > >
> > > For your second question, the optimum flatness is the result of convergence, and does not fully reflect all experiences of parameters during optimization. If the loss landscape for the first layer is totally flat, it would not be necessary to update its parameters at all, right?

---

> > > > ### Public Comment · ~Linara_Adilova1 · 2023-05-23
> > > > **Reply**
> > > >
> > > > Thank you for your answer.
> > > >
> > > > Ok, so the intuition is that while the change in the loss for the shallow layers can be significant, the overall landscape is smoother than for the deep layers? While the deep layers do not have a very smooth landscape and thus require longer to converge, but changing them does not affect overall loss much?

---

### Decision · Program_Chairs · 2023-01-20

**Decision:**

Accept: poster

**Justification For Why Not Higher Score:**

The bottom up convergence has missing prior work: eg https://arxiv.org/abs/1706.05806

**Justification For Why Not Lower Score:**

Quoting: This paper studies the convergence rate during training for different network layers and finds that earlier layers converge more quickly. They also include experiments indicating that the loss landscape is flatter in directions corresponding to early layer weights than late layer weights and suggest that this is connected to the faster convergence and stability. The paper reviewers found these observations compelling and sufficiently novel.

**Metareview: Summary, Strengths And Weaknesses:**

This paper studies the convergence rate during training for different network layers and finds that earlier layers converge more quickly. They also include experiments indicating that the loss landscape is flatter in directions corresponding to early layer weights than late layer weights and suggest that this is connected to the faster convergence and stability. The paper reviewers found these observations compelling and sufficiently novel.

That being said, an essential shortcoming of the manuscript as it stands (and echoing reviewer xk3R) is the lack of a thorough related works section. The observation that early layers converge first and are most stable during training has a long history in the literature. Some references for this point include:

https://arxiv.org/abs/1706.05806 – (Main contribution 2 is bottom up learning of DNNs)
https://arxiv.org/abs/2007.07400 – Find that later layers move the most during task adaptation, while earlier layers are stable

I would strongly recommend modifying the paper to expand section 7 and potentially include this in the introduction, clarifying the distinguishing features of this work!

With this modification I am happy to accept the publication!

**Note From Pc:**

if the above contains the word "oral" or "spotlight" please see: "oral" presentation means -> notable-top-5% and "spotlight" means -> notable-top-25%. As stated in our emails, we are disassociating presentation type from AC recommendations